# Post-translational coordination of chlorophyll biosynthesis and breakdown by BCMs maintains chlorophyll homeostasis during leaf development

Peng Wang [1✉], Andreas S. Richter [1,3], Julius R.W. Kleeberg [2], Stefan Geimer[2] & Bernhard Grimm[1✉]

Chlorophyll is indispensable for life on Earth. Dynamic control of chlorophyll level, determined by the relative rates of chlorophyll anabolism and catabolism, ensures optimal photosynthesis and plant fitness. How plants post-translationally coordinate these two antagonistic pathways during their lifespan remains enigmatic. Here, we show that two *Arabidopsis* paralogs of BALANCE of CHLOROPHYLL METABOLISM (BCM) act as functionally conserved scaffold proteins to regulate the trade-off between chlorophyll synthesis and breakdown. During early leaf development, BCM1 interacts with GENOMES UNCOUPLED 4 to stimulate Mg-chelatase activity, thus optimizing chlorophyll synthesis. Meanwhile, BCM1's interaction with Mg-dechelatase promotes degradation of the latter, thereby preventing chlorophyll degradation. At the onset of leaf senescence, *BCM2* is up-regulated relative to *BCM1*, and plays a conserved role in attenuating chlorophyll degradation. These results support a model in which post-translational regulators promote chlorophyll homeostasis by adjusting the balance between chlorophyll biosynthesis and breakdown during leaf development.

[1] Institute of Biology/Plant Physiology, Humboldt-Universität zu Berlin, Philippstraße 13, 10115 Berlin, Germany. [2] Zellbiologie/Elektronenmikroskopie, Universität Bayreuth, 95440 Bayreuth, Germany. [3] Present address: Institute of Biology/Physiology of Plant Cell Organelles, Humboldt-Universität zu Berlin, Philippstraße 13, 10115 Berlin, Germany. ✉email: wangp2014@gmail.com; bernhard.grimm@rz.hu-berlin.de

During the life cycle of plants, leaf development and seed maturation are usually accompanied by a visible change in pigmentation from pale-yellow to green and then from green to brown or yellow, which reflects the initial rise and subsequent fall in the content of chlorophyll (Chl)[1]. Embedded in the peripheral light-harvesting antenna complexes (LHCs) and core complexes of photosystems I and II (PSI and PSII), Chl absorbs light energy and transfers excitation energy to adjacent pigments to initiate charge separation in the reaction centers of the photosystems[2]. The accumulation of adequate amounts of Chl is therefore vital for plants to establish photosynthetically active chloroplasts during leaf greening[3]. However, free Chl and its metabolic intermediates readily generate singlet oxygen and toxic radicals upon illumination[4,5]. Furthermore, optimized Chl degradation is not only essential for the detoxification of free Chl released from proton synchrotron-large hadron collider (PS-LHC) complexes but also indispensable for the remobilization of nutrients during leaf senescence[6]. Thus, efficient photosynthesis, plant fitness, and grain yield are critically dependent on the dynamic regulation of Chl level in response to various developmental and environmental cues.

The steady-state level of Chl is determined by the relative rates of Chl anabolism and catabolism—processes that largely occur in chloroplasts[7,8]. Chl is synthesized via the magnesium (Mg) branch of tetrapyrrole biosynthesis (TBS)[9,10]. The rate-limiting step in TBS is the formation of 5-aminolevulinic acid (ALA), which is synthesized from glutamyl-tRNA$^{Glu}$ by glutamyl-tRNA reductase (GluTR) and glutamate 1-semialdehyde aminotransferase (GSAT). Six enzymatic steps convert eight molecules of ALA into protoporphyrin IX (Proto), which is the common precursor for both Chl and heme. Mg-chelatase (MgCh), which consists of the catalytic H subunit (CHLH) and the two AAA$^+$ proteins CHLD and CHLI, directs Proto into the Mg branch of TBS by catalysing the insertion of Mg$^{2+}$ into Proto to generate Mg-Proto (MgP)[11,12]. Remarkably, the GENOMES UNCOUPLED 4 (GUN4) is capable of binding Proto and MgP and stimulates MgCh activity[13–15]. MgP is further converted into MgP monomethylester (MgPMME) and protochlorophyllide (Pchlide) by MgP methyltransferase (CHLM)[16] and MgPMME cyclase[17], respectively. In angiosperms, Pchlide is reduced to chlorophyllide (Chlide) exclusively by the light-dependent NADPH:Pchlide oxidoreductase (POR)[18]. Subsequently, Chl synthase (CHLG) esterifies Chlide with a phytol chain to form the hydrophobic Chl $a$[19], which can be further converted into Chl $b$ by Chlide $a$ oxygenase (CAO)[20]. Finally, newly synthesized Chl $a$ and Chl $b$ are rapidly integrated into the Chl-binding proteins of PS-LHC complexes[21].

As a visible symptom of leaf senescence and fruit ripening, Chl breakdown is mediated by the pheophorbide $a$ oxygenase (PAO)/ phyllobilin pathway[22], which is initiated by conversion of Chl $b$ into Chl $a$ by the combined action of NON-YELLOW COLORING1 (NYC1)[23,24], NYC1-LIKE (NOL), and 7-hydroxymethyl Chl $a$ reductase (HCAR)[25]. Mg-dechelatase, encoded by the NON-YELLOWING/STAY-GREEN (NYEs/SGRs) genes[26,27], is the first committed enzyme in the PAO/phyllobilin pathway and removes Mg$^{2+}$ from Chl $a$ to form pheophytin $a$ (Phein $a$), which is then hydrolyzed to Pheide $a$ and a phytol chain by PHEO-PHYTINASE (PPH)[28]. PAO cleaves the porphyrin ring of Pheide $a$ to generate an oxidized red Chl catabolite (RCC)[29], which is subsequently acted upon by RCC reductase to produce a primary fluorescent Chl catabolite ($p$FCC). Finally, $p$FCC is modified, transported into the vacuole, and isomerized to a non-fluorescent product[30]. Intriguingly, genetic lesions affecting early steps in Chl catabolism lead to a cosmetic stay-green trait[31], a syndrome in which—by definition—leaf senescence proceeds normally, but Chl and pigmented Chl catabolites, such as Phein $a$ and Pheide $a$, are retained.

Chl homeostasis requires constant adjustment of rates of Chl biosynthesis and breakdown to prevent mutual neutralization of their metabolic activities. In support of this notion, the half-life of Chl was determined to be as short as 6 to 8 h at the beginning of de-etiolation of cereal leaves, but increases to 50 h when leaf greening is complete[32]. However, the mechanisms underlying the balance between Chl synthesis and breakdown during the lifespan of plants remain largely unknown. At the transcriptional level, Chl biosynthesis genes (CBGs) are preferentially up-regulated and Chl catabolic genes (CCGs) are suppressed during leaf greening, and this pattern is reversed in the course of senescence[33,34]. Several transcriptional factors modulate the relative levels of Chl biosynthesis and catabolism. In Arabidopsis, LONG HYPOCO-TYL 5 and GOLDEN-2-LIKE (GLK2) activate expression of key CBGs in response to light signaling[35,36]. Conversely, PHYTO-CHROME B and EARLY FLOWERING 3 suppress expression of CCGs by inhibiting the senescence-promoting activity of PHYTOCHROME-INTERACTING FACTOR 4 and 5[37]. Nevertheless, it is still puzzling how plants maintain stable Chl levels during young leaf development when expression of the CCGs is being progressively up-regulated[38].

The interplay of the two antagonistic Chl metabolic pathways in the context of early leaf growth and senescence has long been discussed[31,32]. However, no evidence exists for a post-translational connection between the two pathways mediated either by a common regulator or by two interacting enzymatic steps, which simultaneously sense the need for Chl synthesis or degradation[39]. It is hypothesized that plants have evolved as yet unknown post-translational regulatory mechanisms to stimulate Chl synthesis and inhibit Chl degradation, so as to maintain Chl homeostasis during leaf development. To test this hypothesis, we performed a reverse genetic screen for genes that might function in both Chl synthesis and catabolism, and identified two paralogous genes in Arabidopsis, which we name BALANCE of CHLOROPHYLL METABOLISM 1 and 2 (BCM 1 and 2). We present evidence that BCM1 and BCM2 play highly conserved roles in both Chl metabolic pathways. While BCM1 is the predominant isoform during seedling growth, BCM2 exerts its control over Chl amounts during leaf senescence. Thus, we propose that the fine-tuning of dynamically changing Chl levels from leaf emergence to senescence involves post-translational coordination of Chl synthesis and breakdown, mediated by conserved auxiliary factors.

## Results

**Identification of BCM1 in Arabidopsis.** To identify novel regulators of Chl biosynthesis and catabolism, we screened publicly available datasets for Arabidopsis genes of unknown function that exhibit the transcriptional signatures of CBGs or CCGs, and identified At2g35260, hereafter designated BCM1 (because of its dual function in both Chl metabolic pathways, see below). The BCM1 transcript clusters with key CBGs, showing an expression pattern that is most similar to that of GUN4 (Fig. 1a). Immunoblot analyses using a BCM1 antiserum raised against recombinant Arabidopsis His-BCM155 showed that BCM1 accumulates as an ~36 kDa protein in all Arabidopsis tissues except roots (Fig. 1b). The highest levels of BCM1 and Chl biosynthesis enzymes (CBEs) were observed in young and mature rosette leaves, and dramatically decreased during senescence. Moreover, trace amounts of BCM1 accumulated in etiolated seedlings and rapidly increased upon illumination, as do CBEs and proteins of the photosynthetic apparatus (Fig. 1c).

BCM1 encodes a 382-amino-acid protein with an N-terminal chloroplast transit peptide (cTP) and six transmembrane domains (TMDs) (Fig. 1d). Transient expression of BCM1 fused to yellow fluorescent protein (YFP) in Arabidopsis protoplasts

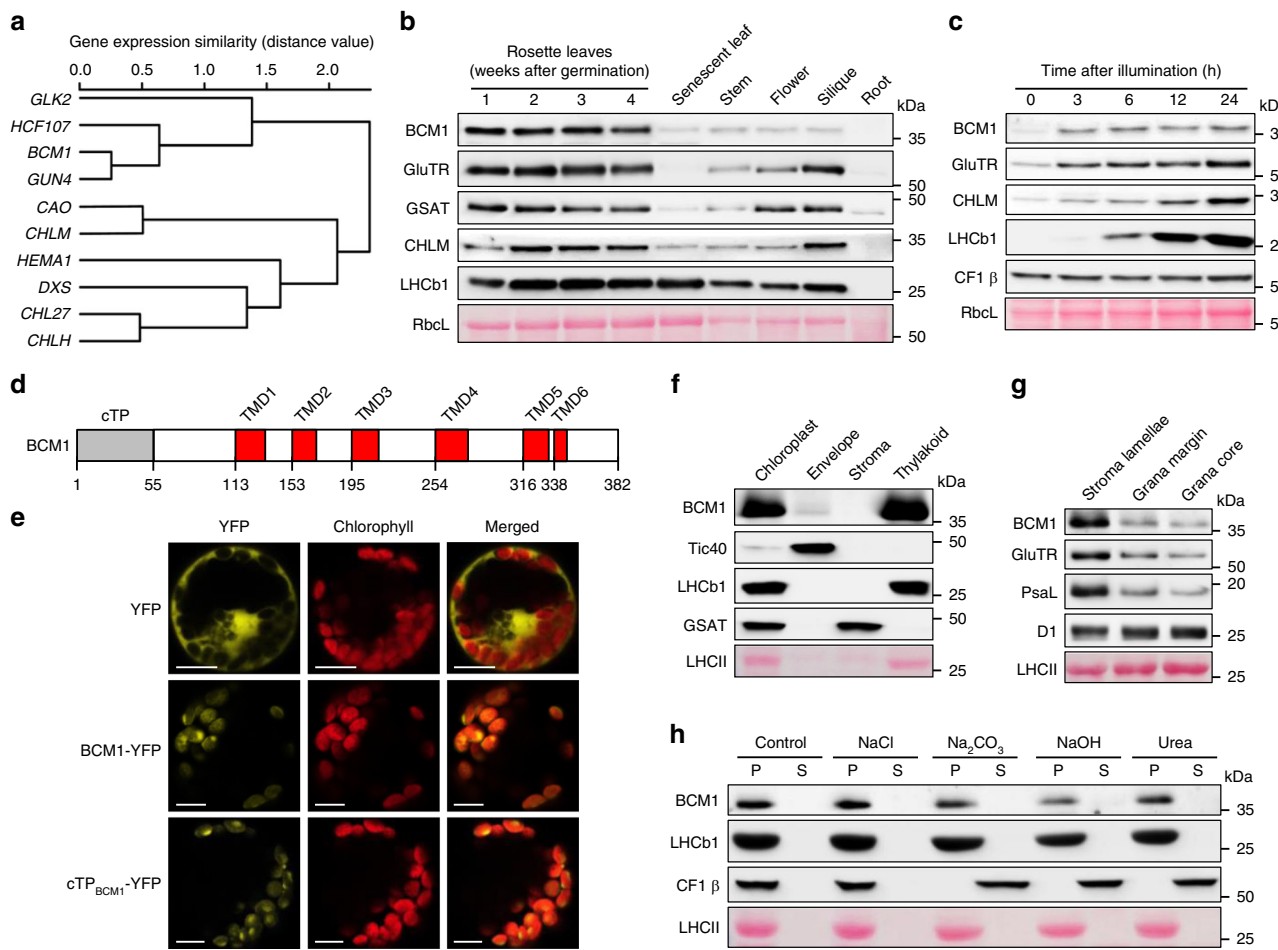

**Fig. 1 Characterization of BCM1. a** Co-expression analysis of *BCM1*. *BCM1* together with the *CBG*s, *GLK2*, the carotenoid biosynthesis gene *DEOXYXYLULOSE-5-PHOSPHATE SYNTHASE* (*DXS*), and the gene encoding the chloroplast protein *HIGH CHLOROPHYLL FLUORESCENT 107* (*HCF107*) were hierarchically clustered based on pairwise levels of similarity between their expression profiles, using the gene co-expression database ATTEDII (http://atted.jp/)[76]. Low distance values indicate high degrees of co-expression. **b** Accumulation of BCM1, CBEs, and LHCb1 in different organs of *Arabidopsis*. **c** Light-induced accumulation of BCM1, CBEs, and LHCb1 in 5-day-old etiolated WT seedlings, following illumination (80 μmol photons m$^{-2}$ s$^{-1}$) for 0, 3, 6, 12, and 24 h. **d** Schematic overview of the domain structure of BCM1. The cTP is shown in gray, while the six predicted TMDs are shown in red. **e** Subcellular localization of the BCM1-YFP fusion protein. Both BCM1-YFP and cTP$_{BCM1}$-YFP fluorescence coincides with Chl autofluorescence, confirming chloroplast targeting of BCM1. In the control, YFP itself accumulates in the cytosol and nucleus. Scale bars, 5 μm. **f** Suborganellar localization of BCM1. Chloroplasts from WT seedlings were subfractionated into envelope, stroma, and thylakoid fractions. For comparison, the proteins TRANSLOCON AT THE INNER ENVELOPE MEMBRANE OF CHLOROPLAST 40 (Tic40), GSAT, and LHC were specifically located in the envelope, stroma, and thylakoid, respectively. **g** Distribution of BCM1 across the different thylakoid membrane regions. Thylakoid proteins were solubilized with digitonin and fractionated into grana core, grana margins, and stroma lamellae. **h** Salt washing of thylakoid membranes. The WT thylakoids were sonicated in the presence of 1 M NaCl, 200 mM Na$_2$CO$_3$, 0.1 M NaOH, and 6 M urea on ice for 30 min before centrifugation to separate membrane (P) from soluble (S) fractions. Untreated thylakoids served as the control. LHCb1 and the CF1 β subunit of ATP synthase, representing an intrinsic membrane protein and a peripheral thylakoid-associated protein, respectively, were used as positive controls for binding affinity to thylakoid membrane fractions. In **b**, **c**, **f–h**, immunoblot analyses were conducted using the indicated antibodies. Ponceau S-stained membrane strips bearing the large subunit of Rubisco (RbcL) or LHC proteins of PSII (LHCII) were used as loading controls.

reveals chloroplast localization for BCM1 (Fig. 1e). Immunoblot analyses of isolated envelope, stroma, and thylakoid fractions of chloroplast showed that ~92% of BCM1 was located in the thylakoid membrane and only ~8% in the envelope fraction (Fig. 1f). The thylakoid membrane is organized into grana stacks and stroma lamellae. Most known proteins involved in the biogenesis and maintenance of the photosynthetic apparatus in the thylakoids, including Chl catabolism, are predominantly located in the stroma lamellae[3,22,40]. We found that BCM1, GluTR, and a PSI subunit (PsaL) are clearly enriched in the stroma lamellae, and to a lesser degree in the grana margins and grana stacks (Fig. 1g). To clarify whether the BCM1 acts as an integral or peripheral thylakoid protein, isolated thylakoids were

treated with chaotropic and alkaline reagents to release membrane-associated proteins. BCM1 behaved like the integral LHC proteins (with three TMDs), which were resistant to all of the treatments applied (Fig. 1h). Therefore, BCM1 is an intrinsic membrane protein, and is mainly localized in the non-appressed regions of the thylakoid membrane.

**BCM1 is required for efficient Chl biosynthesis**. BCM1's ortholog in soybean (*Glycine max L. Merr.*) has been shown to be encoded by the stay-green *G* gene and to play a conserved function in controlling seed dormancy in soybean, rice, and *Arabidopsis*[41,42]. However, its function in Chl metabolism remains to be addressed.

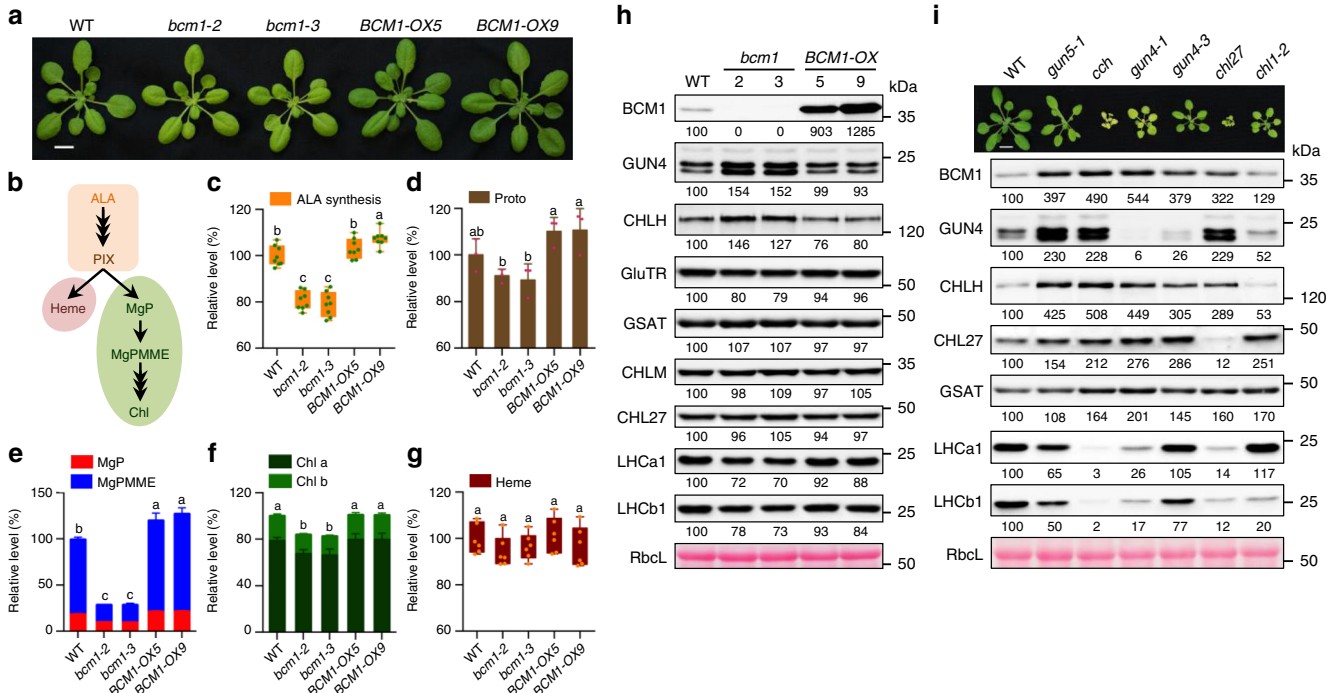

**Fig. 2 BCM1 is a positive regulator of Chl biosynthesis in *Arabidopsis*. a** Representative images of 28-day-old WT, *bcm1* and *BCM1-OX* seedlings grown under short-day normal light (120 μmol photon m$^{-2}$ s$^{-1}$) conditions. Scale bar, 1 cm. **b** Scheme of TBS in higher plants. **c–g** Relative ALA synthesis rate (**c**), Proto (**d**), Mg-porphyrin (**e**), Chl (**f**), and heme (**g**) levels in 18-day-old WT, *bcm1*, and *BCM1-OX* seedlings grown under the same conditions as in **a**. Error bars represent SD of eight, three, four, twelve, and six biological replicates, respectively. Letters above histograms indicate significant differences as determined by Tukey's HSD method ($P < 0.05$). **h** Steady-state levels of the indicated proteins in seedlings analyzed in **a** were determined by immunoblot analysis using the indicated antibodies. **i** Steady-state levels of the indicated proteins in 28-day-old WT, *gun5-1*, *cch*, *gun4-1*, *gun4-3*, *chl27*, and *ch1-2* seedlings, grown under short-day low light (70 μmol photons m$^{-2}$ s$^{-1}$) conditions were quantified by immunoblotting using the indicated antibodies. A representative image of the *Arabidopsis* seedlings analyzed in **i** is shown. In **h**, **i**, Ponceau S-stained membrane strips bearing RbcL were used as loading controls. Numbers below immunoblots represent normalized protein abundances relative to WT seedlings.

To investigate the function of BCM1 in Chl biosynthesis, we characterized the status of TBS in seedlings of *bcm1* mutants and *BCM1*-overexpressing (*BCM1-OX*) plants (Fig. 2a). We identified three allelic *Arabidopsis bcm1*-null mutants, which showed a pale-green leaf phenotype both under standard growth conditions and when exposed to various light-stress regimes (Fig. 2a and Supplementary Figs. 1a–c, 2). These findings suggest that BCM1 is important for accurate Chl accumulation under varying light intensities and photoperiods. Indeed, ALA synthesis, which is the rate-limiting step of TBS, was decreased in *bcm1* mutants (Fig. 2b, c), and supplementation with ALA failed to rescue the pale-green leaf phenotype (Supplementary Fig. 3). Reduced ALA synthesis in *bcm1* led to slightly reduced accumulation of Proto (Fig. 2d). Markedly reduced flux of Mg-porphyrins (including MgP and MgPMME) through the Mg branch of TBS, and reduced Chl contents, were correspondingly observed in *bcm1* mutants (Fig. 2e, f). In contrast, lack of BCM1 did not affect the accumulation of non-covalently bound heme (Fig. 2g). In comparison to wild-type (WT) seedlings, *BCM1-OX* seedlings showed significantly increased ALA synthesis rates and elevated Proto and Mg-porphyrin levels, as well as WT-like contents of Chl and heme (Fig. 2c–g). Similar stimulatory effects on the ALA synthesis rate, and increased accumulation of Mg-porphyrins, have been observed in *GUN4*, *CHLH*, and *CHLM* overexpression lines[14,43,44]. Taken together, these results suggest a positive role for BCM1 in the Mg branch of the TBS pathway.

To further elucidate the molecular function of BCM1 in Chl biosynthesis, the expression of *CBG*s and steady-state levels of CBEs were examined in *bcm1* and *BCM1-OX* seedlings. Immunoblot analyses showed that GluTR levels in *bcm1* mutants were

slightly decreased (to ~80% of the WT value), whereas GUN4 and CHLH contents were substantially increased to ~153% and ~137% of those in WT, respectively (Fig. 2h). In addition, reduced Chl levels in *bcm1* correlated with decreased levels of LHCa1 and LHCb1 (representative LHC proteins of PSI and PSII), in agreement with the highly synchronized synthesis of Chls and LHC proteins[21,45]. In contrast to these perturbations in *bcm1*, *BCM1-OX* seedlings accumulated WT-like levels of CBGs and LHC proteins (Fig. 2h), which excludes the possibility that BCM1 is implicated in the proteolysis of CBEs. The quantitative real-time PCR (qRT-PCR) analyses showed that transcripts of all *CBG*s attained WT-like levels in both *bcm1* and *BCM1-OX* seedlings (Supplementary Fig. 4a). Thus, BCM1 is suggested to act post-translationally on Chl biosynthesis.

Strikingly, Chl synthesis mutants (Supplementary Table 1), including *gun5-1* and *cch* (two missense mutants of CHLH[46]), *gun4-1* and *gun4-3* (two knockdown mutants of the MgCh regulator GUN4, with GUN4 being less stable in *gun4-1* than in *gun4-3*[13–15]), and *chl27* (a knockdown mutant of MgPMME cyclase), contained increased levels of BCM1, GUN4, and CHLH in comparison to WT, whereas *ch1-2* (a missense mutant of CAO) accumulated WT-like level of BCM1 and reduced contents of GUN4 and CHLH (Fig. 2i). Moreover, the elevated levels of BCM1, GUN4, and CHLH seen in *chlh*, *gun4*, and *chl27* mutants could not be attributed to up-regulated expression of the corresponding transcripts (Supplementary Fig. 4b). These results reveal a post-translational correlation between BCM1, GUN4, and CHLH, in that dysfunction of MgCh or MgPMME cyclase is associated with increased stability of BCM1, GUN4, and CHLH.

**BCM1 contributes to the control of the Mg chelation step**. The drastically reduced amounts of Mg-porphyrins and increased GUN4 and CHLH levels observed in *bcm1* (Fig. 2e, h) phenotypically resemble the effects of impaired Chl biosynthesis in MgCh mutants, such as missense or knockdown mutants of *CHLH* and *GUN4*[13,15,46]. Therefore, the genetic interaction between BCM1 and MgCh was investigated by combining *bcm1-3* with *gun5* and with two allelic *gun4* mutants (Fig. 3a). The double mutants accumulated steady-state protein levels of CBEs comparable to those found in the single mutants (Fig. 3b). Intriguingly, BCM1 deficiency exacerbated the retarded growth phenotypes of the MgCh single mutants, but had varying effects on leaf pigmentation. Relative to the single mutants, the *bcm1-3 gun5-1* double mutant exhibited a more pronounced reduction in the rate of ALA synthesis and greater loss of Chl (in particular along the leaf veins) and Mg-porphyrins (Fig. 3c, d). These results reveal a synergistic effect of the *bcm1* and *gun5-1* mutations on MgCh

activity, suggesting that BCM1 and CHLH play distinct roles in promoting MgCh activity. However, *bcm1-3 gun4-1* and *bcm1-3 gun4-3* seedlings mimic the defects in Chl biosynthesis observed in *gun4-1* seedlings, including decreased ALA synthesis activity and reduced Chl and Mg-porphyrin contents (Fig. 3c, d). These data indicate that GUN4 acts downstream of BCM1.

To study the epistatic relationships between BCM1, CHLH/GUN5, and GUN4 further, we crossed *BCM1-OX* plants with *gun5-1* and two *gun4* mutants. We found that overproduction of BCM1 in the *gun5-1* and *gun4* mutant background did not alter the steady-state levels of CBEs or LHC proteins (Fig. 3f). Unlike *BCM1-OX* seedlings, *BCM1-OX/gun5-1*, *BCM1-OX/gun4-1*, and *BCM1-OX/gun4-3* seedlings did not show elevated levels of Mg-porphyrins relative to the either *gun5-1* or the *gun4* mutants (Fig. 3g, h). We therefore conclude that BCM1-mediated promotion of MgCh activity does not compensate mutant CHLH and critically relies on the action of GUN4.

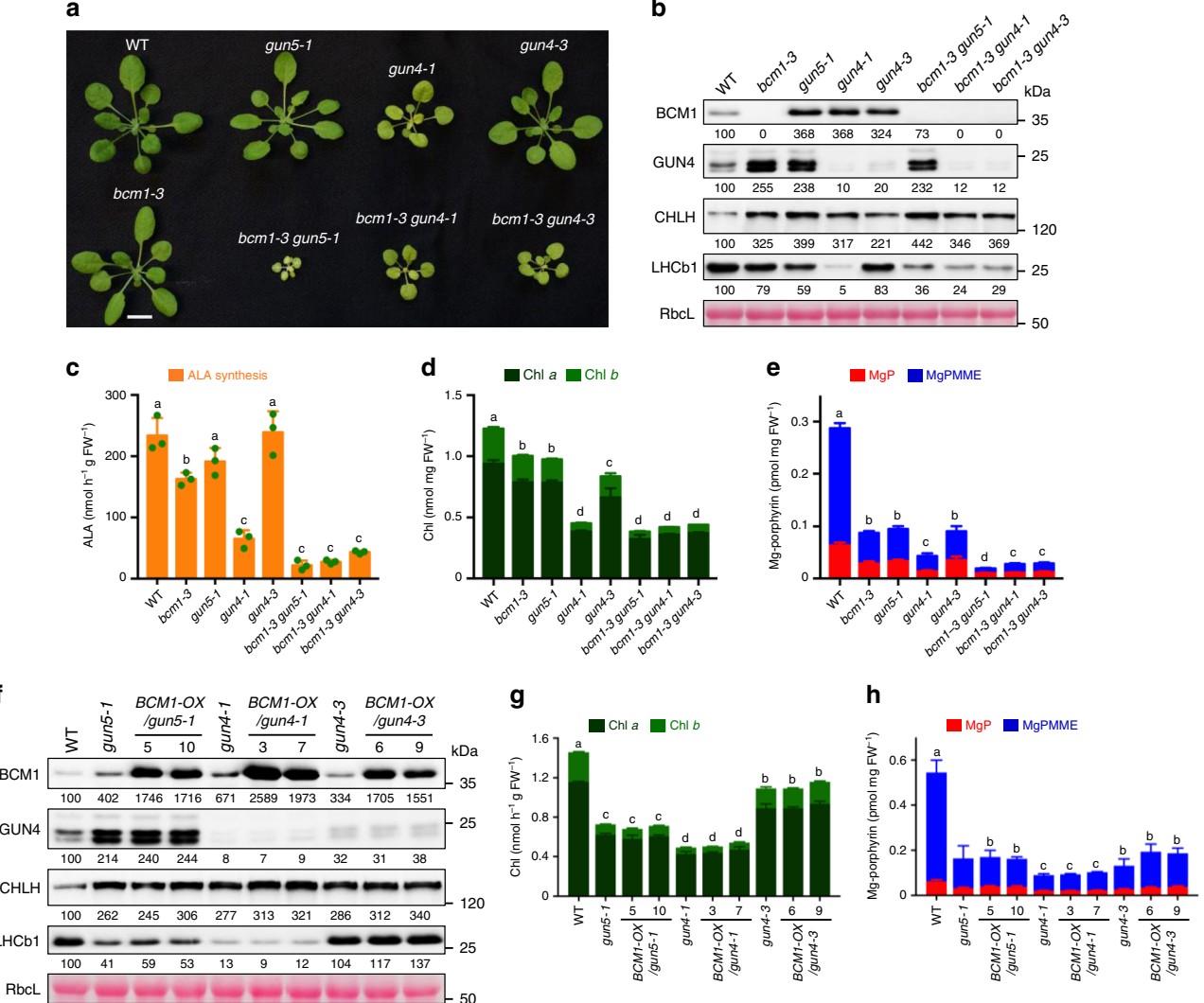

**Fig. 3 Genetic interaction between BCM1 and MgCh. a** Representative images of 28-day-old WT, *bcm1-3*, *gun5-1*, *gun4-1*, *gun4-3*, *bcm1-3 gun5-1*, *bcm1-3 gun4-1*, and *bcm1-3 gun4-3* seedlings grown under short-day low light (70 μmol photon m⁻² s⁻¹) conditions. Scale bar, 1 cm. **b** Steady-state levels of the indicated proteins in the seedlings depicted in **a** were determined by immunoblotting using the indicated antibodies. **c–e** ALA synthesis rate (**c**) and levels of Chl (**d**) and Mg-porphyrin (**e**) in the seedlings shown in **a**. **f** Steady-state levels of the indicated proteins in 21-day-old WT, *gun5-1*, *BCM1-OX/gun5-1*, *gun4-1*, *BCM1-OX/gun4-1*, *gun4-3*, and *BCM1-OX/gun4-3* seedlings grown under the same conditions as in **a** were determined by immunoblotting using the indicated antibodies. **g**, **h** Levels of Chl (**g**) and Mg-porphyrins (**h**) in seedlings analyzed in **f**. In **c–e**, **g**, **h**, error bars represent SD of three biological replicates. Letters above histograms indicate significant differences determined by Tukey's HSD method (P < 0.05). In **b**, **f**, Ponceau S-stained membrane strips bearing RbcL were used as loading controls. Numbers below immunoblots represent normalized protein abundances relative to WT seedlings.

Since increased GUN4 and CHLH levels were observed in the *chl27*-knockdown mutant (Fig. 2i), we examined the possibility that BCM1 also contributes to the MgPMME cyclase step by comparing the molecular phenotypes of *bcm1-3*, *chl27*, and *bcm1-3 chl27* mutants (Supplementary Fig. 5a, b). While *bcm1-3* showed drastically reduced contents of both MgP and MgPMME, *chl27* mutants accumulated 8-fold more MgPMME than WT, owing to the impaired conversion of MgP to MgPMME catalyzed by CHL27. In comparison with the *chl27* single mutant, *bcm1-3 chl27* plants formed rosette leaves of increased size, and levels of MgPMME were greatly reduced (Supplementary Fig. 5c, d). Thus, these data suggest that BCM1's function in Chl biosynthesis acts on an enzymatic step that lies upstream of the MgPMME cyclase. Elevated growth of *bcm1-3 chl27* seedlings relative to *chl27* is explained by reduced photosensitization due to lower MgPMME content.

**BCM1 stimulates MgCh activity via its interaction with GUN4**. A yeast two-hybrid (Y2H) screen for potential partners of BCM1 in the TBS pathway detected interactions with GluTR, GUN4, and CHLM (Fig. 4a). These interactions were confirmed by the bimolecular fluorescence complementation (BiFC) approach (Fig. 4b). The physical interaction between BCM1 and GUN4 is compatible with the genetic interaction between them (Fig. 3). To determine whether this interaction is necessary for the stimulation of MgCh activity, the effect of BCM1 on MgCh activity was explored in vitro using recombinant MgCh components[15,47]. To exclude an effect of endogenous GUN4 on the stimulation of

MgCh activity, we isolated thylakoid membranes from *bcm1-3 gun4-1* and *BCM1-OX3/gun4-1* seedlings. The recombinant MgCh subunits were then incubated with thylakoid membranes in the presence or absence of recombinant His-GUN4. When His-GUN4 was omitted, the addition of isolated thylakoid membranes had no effect on MgCh activity (Fig. 4c). In accordance with previous reports[15,47], addition of His-GUN4 greatly stimulated MgCh activity. Notably, the addition of *bcm1-3 gun4-1* thylakoids significantly diminished MgCh activity compared to the assay without thylakoids; conversely, the use of thylakoids from *BCM1-OX3/gun4-1* increased MgCh activity by ~12% compared to the assay supplemented with thylakoids from *bcm1-3 gun4-1* seedlings (Fig. 4c).

To further prove these observations, we expressed His-BCM1 in the membranes of *Saccharomyces cerevisiae* cells and conducted MgCh assay by using isolated yeast membranes. We found the BCM1-containing membranes exhibited a ~12% higher activity than control membranes (without BCM1) in the presence of His-GUN4 and the three MgCh subunits (Supplementary Fig. 6a). Next, we analyzed the effect of purified His-BCM1 on MgCh activity in vitro. Because BCM1 is an integral membrane protein, the purification of recombinant BCM1 proteins required supplementation of the buffer with a non-ionic detergent, such as *n*-dodecyl-β-D-maltoside (β-DM). However, the addition of 0.15 mM β-DM (its critical micelle concentration) essentially abolished MgCh activity (Supplementary Fig. 6b). In this context, the use of 1 μM His-BCM1 can still increase MgCh activity by ~3-fold compared with the assays without BCM1 or supplemented with 1 μM glutathione-*S*-transferase (GST) (Supplementary Fig. 6c). Taken together, these

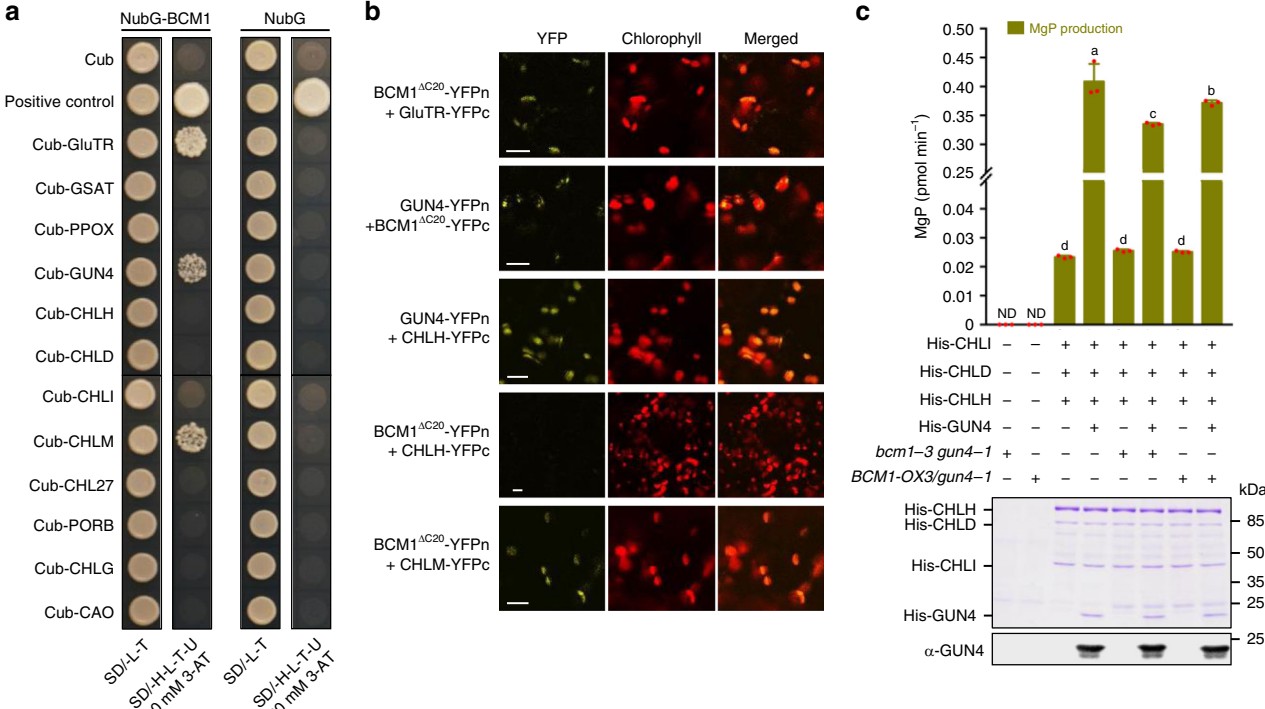

**Fig. 4 BCM1 interacts with GUN4 and stimulates MgCh activity via GUN4. a** Y2H analyses of interactions between BCM1 and CBEs. The transformed yeast strains were analyzed on selective medium lacking Leu and Trp (SD/-L-T) or His, Leu, Trp and Ura (SD/-H-L-T-U) in the presence of 30 mM 3-amino-1,2,4-triazole (3-AT). The combination of NubG-GluTR and Cub-GBP was used as the positive control. The NubG was used as the negative control for NubG-BCM1. **b** BiFC assays for interactions between BCM1 and CBEs. Co-expression of BCM1$^{\Delta C20}$-YFPn with CHLH-YFPc served as the negative control, and the combination of GUN4-YFPn with CHLH-YFPc served as the positive control. Scale bars, 10 μm. **c** In vitro MgCh assay. Production of MgP in the assay was measured by HPLC and quantified relative to incubation time. Recombinant proteins used in the assay were stained with Coomassie Brilliant Blue (middle panel) and probed with GUN4 antibody (bottom panel), respectively. ND, not detected. Error bars represent SD of three biological replicates. Letters above histograms indicate significant differences as determined by Tukey's HSD method ($P < 0.05$).

results thus substantiate the assumption that BCM1 interacts with GUN4 and further stimulates MgCh activity at the thylakoid membrane.

Reduction in MgCh activity in planta was always accompanied by a lower capacity for ALA synthesis (Fig. 3c)[15,46], suggesting the existence of an essential feedback mechanism to prevent the accumulation of phototoxic Proto when MgCh is blocked[7,10,39]. We therefore propose that BCM1 and GUN4 function at the interface between ALA synthesis and the Mg chelation of Proto. This hypothesis is supported by the fact that overexpression of either *BCM1* or *GUN4* stimulates ALA synthesis (Fig. 2c)[14] and by the finding that BCM1 physically interacts with GUN4 and GluTR (Fig. 4a, b). However, further characterization of the physical interaction of GUN4 with GluTR and its implications for the control of ALA synthesis in the presence or absence of BCM1 will be needed to substantiate this hypothesis.

Although BCM1 also interacts with CHLM, neither the stability nor the activity of CHLM is altered in *bcm1* seedlings (Fig. 2h and Supplementary Fig. 7). Furthermore, unlike *bcm1* mutants, *chlm* plants accumulate excess MgP (the substrate for CHLM) relative to WT[16]. Therefore, it is unlikely that the reduced Chl biosynthesis in *bcm1* is attributable to altered function of CHLM.

**Conserved role of two BCM paralogs in Chl biosynthesis**. BCM1 orthologs are broadly conserved in land plants, but are not found in green algae or cyanobacteria (Supplementary Fig. 8). *BCM1* has a paralog in *Arabidopsis*, *At*4g17840, which we term *BCM2*. It encodes a 422-amino-acid protein with an extended N-terminal sequence relative to BCM1. The two BCMs share 87% sequence similarity and 75% sequence identity (Supplementary Fig. 9a). While *BCM1* and some *CBG*s show a similar expression pattern, *BCM2*'s expression resembles those of key *CCG*s, such as *SGR1* and *PAO*, which are highly expressed in senescent leaves and dry seeds (Supplementary Fig. 9b, c). Thus, BCM2 is expected to act in late developmental stages only.

BCMs have been annotated as CAAX-type endopeptidases[42], which are involved in protein isoprenylation[48]. So far, two classic types of CAAX peptidase, the zinc metallopeptidase STE24 and the RAS-CONVERTING ENZYME 1 (RCE1), have been described in *Arabidopsis*[49,50]. In contrast to both *BCM*s, expression of either *STE24* or *RCE1* was able to rescue a CAAX peptidase-deficient yeast mutant strain (*rce1Δ ste24Δ*) (Supplementary Fig. 10a). These results strongly suggest that BCMs do not exhibit CAAX protease activity in vivo. As the polyclonal BCM1 antibody was raised against the highly conserved N-terminal sequence of the protein 55–253, the antiserum also recognizes BCM2 on immunoblots (Supplementary Fig. 10b).

Next, we quantified the contributions of the two BCM paralogs to the optimization of Chl biosynthesis using *bcm2* T-DNA insertion knockdown mutants and the homozygous *bcm1-3 bcm2-2* double mutant (Supplementary Figs. 1d–g and 11a, b). We found that *bcm2* seedlings showed a WT-like phenotype. In agreement with this observation, the *bcm1-3* single mutant and the *bcm1-3 bcm2-2* double mutant exhibited very similar Chl biosynthesis phenotypes, including reduced contents of Chl and Mg-porphyrin, increased levels of GUN4 and CHLH, and WT-like amounts of heme (Supplementary Fig. 11c–e). Furthermore, the efficient gene silencing of *BCM2* was achieved by a virus-induced gene silencing (VIGS) approach[51] in WT or *bcm1-3* plants (Fig. 5a, b). The *BCM2* silencing (VIGS-BCM2/WT) seedlings also showed WT-like phenotype; however, inactivation of both *BCMs*' expression (VIGS-BCM2/*bcm1-3*) led to distinct leaf pigmentation in mature leaves (Fig. 5a, c, d). In VIGS-BCM2/*bcm1-3* seedlings, young leaves had similar levels of Chl as *bcm1* (VIGS-GFP/*bcm1-3*), while old leaves displayed an enhanced pale-green leaf phenotype

compared to those of VIGS-GFP/*bcm1-3* seedlings. Consistently, steady-state levels of LHC proteins were greatly reduced in old leaves of VIGS-BCM2/*bcm1-3* seedlings (Fig. 5e). In contrast, both VIGS-GFP/*bcm1-3* and VIGS-BCM2/*bcm1-3* plants accumulated the similar amounts of TBS proteins, such as GUN4, CHLH, and CHLI (Fig. 5e). However, expressions of senescence-associated gene (*SAG*), such as *SAG12*, and *SGR1* were similar in VIGS-BCM2/*bcm1-3* seedlings as that in VIGS-GFP/WT and VIGS-GFP/*bcm1-3* seedlings (Supplementary Fig. 12). These data suggest a predominant role for BCM1 in the regulation of Chl biosynthesis and a need of two BCM isoforms for precise control of amount of Chl in adult leaves.

Because *BCM2* is highly expressed in senescent leaves, it is plausible that its low expression level in young seedlings is insufficient to compensate for BCM1 deficiency. To verify this idea, we generated the *bcm1-2* complementation lines that constitutively expressed *BCM2*, driven by the 35S promoter (*BCM2-OX*/*bcm1*) (Fig. 5f, g). Overexpression of *BCM2* indeed led to the accumulation of BCM2 in planta (as detected by immunoblot analyses) and completely abrogated the negative effect of the *bcm1* mutation on Chl biosynthesis, restoring Chl and Mg-porphyrin contents to normal and suppressing the increases in GUN4 and CHLH levels (Fig. 5h–j). Moreover, BCM2 interacts with the same proteins of Chl biosynthesis (Supplementary Fig. S13). Hence, BCM2 can functionally substitute for BCM1 and thus retains a positive function in Chl biosynthesis.

**Both BCMs disturb Chl breakdown**. The up-regulated expression of *BCM2* and key *CCG*s during senescence (Supplementary Fig. 9b, c) and enhanced pale-green leaf phenotype in old leaves of VIGS-BCM2/*bcm1-3* plants (Fig. 5a, c) prompted us to investigate the potential function of the two *Arabidopsis* BCM paralogs in Chl catabolism. To this end, Chl degradation rates were determined in leaves detached from 5-week-old WT, *bcm1* and *BCM1-OX* seedlings and subjected to dark-induced senescence (DIS). After 7 days of dark incubation (DDI), detached *bcm1* leaves were markedly paler, with less Chl than the WT, while Chl degradation was delayed in detached leaves of *BCM1-OX* seedlings, in which levels of both Chl and the Chl catabolite Phein *a* were higher than in WT (Supplementary Fig. 14a–c). In accordance with these observations, the PS-LHC complexes were more stable in *BCM1-OX* than in WT seedlings after 7 DDI (Supplementary Fig. 15), while *bcm1* seedlings showed the opposite phenotype, with a complete breakdown of PS-LHC complexes. In contrast to the stable PS-LHC complexes, cytochrome $b_6f$ and ATP synthase complexes were degraded in *BCM1-OX* seedlings during DIS, as they were in WT and *bcm1* seedlings under the same conditions (Supplementary Fig. 15). As essential components of the leaf senescence syndrome, increased ion leakage rates (Supplementary Fig. 14d) and up-regulation of *CCG*s (Supplementary Fig. 14e) were comparatively determined in WT, *bcm1* and *BCM1-OX* seedlings during dark incubation, suggesting that overexpression of *BCM1* specifically interfere with the rate of Chl breakdown during DIS. In summary, these results suggest that *BCM1-OX* seedlings show a cosmetic stay-green phenotype during long-term dark incubation.

To investigate the contribution of BCM2 to the control of Chl breakdown, we compared Chl degradation rates in the WT, *bcm1-3 bcm2-2*, *BCM1-OX9*, and *BCM2-OX1*/*bcm1* seedlings during DIS. We found enhanced Chl breakdown in *bcm1-3 bcm2-2* double mutant compared to WT during dark incubation, as indicated by the far less retained Chl content in *bcm1-3 bcm2-2* than in WT at 7 DDI (Fig. 6a–c). Like *BCM1-OX* plants, *BCM2-OX1*/*bcm1* seedlings showed a stay-green phenotype at 7 DDI (Fig. 6a–c). The

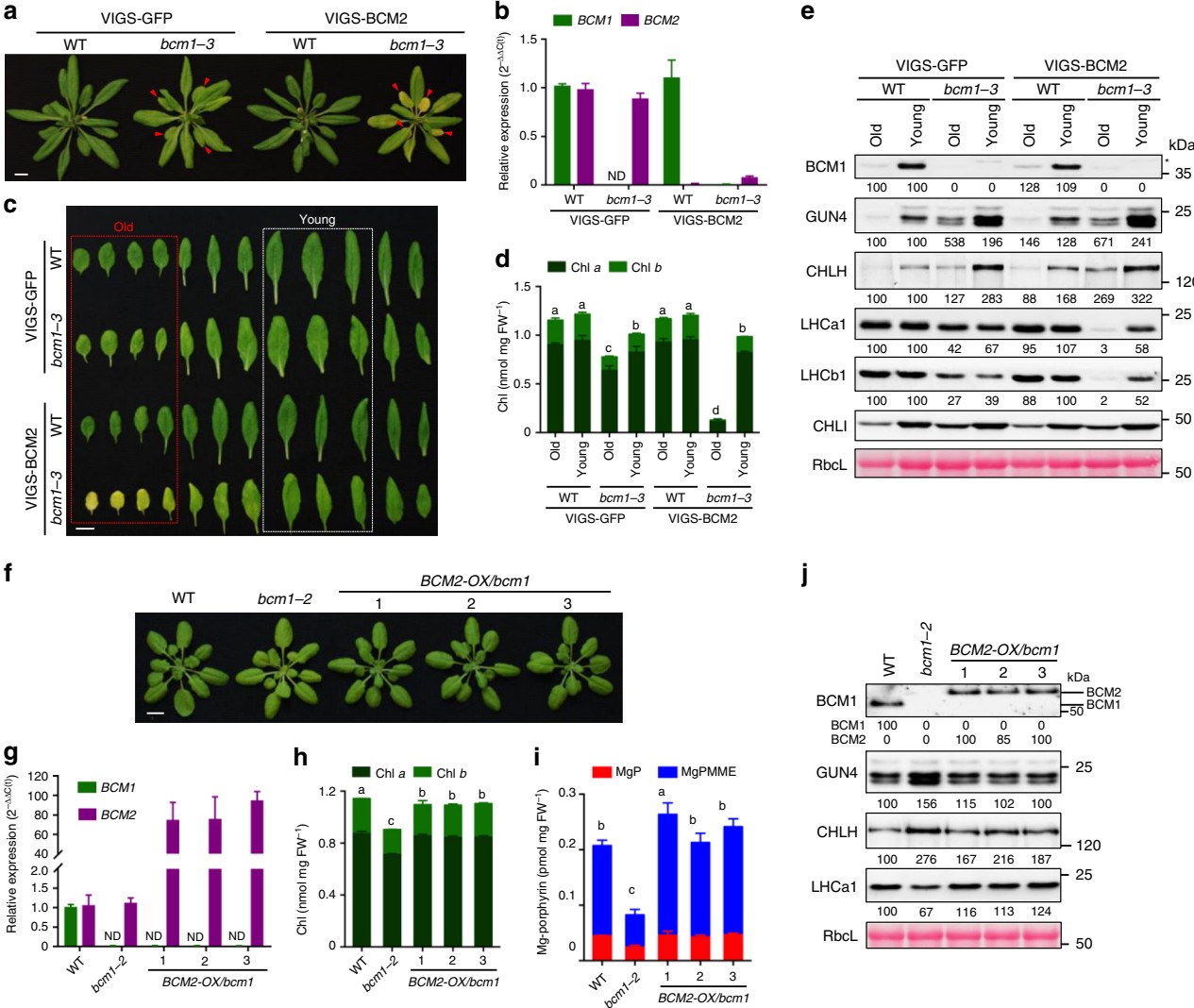

**Fig. 5 BCM2 plays a conserved role in Chl biosynthesis. a** Representative images of 35-day-old VIGS-GFP (as the negative control) and VIGS-BCM2 seedlings, which were infected with the pTRV2-GFP and pTRV2-BCM2 vectors via agroinfiltration in the WT and *bcm1-3* background, respectively. The VIGS seedlings were grown under long-day normal light (120 μmol photon m$^{-2}$ s$^{-1}$) conditions. The old leaves were indicated by red arrows. **b** qRT-PCR analysis of *BCM1* and *BCM2* transcripts in the seedlings shown in **a**. **c** Representative images of detached leaves from seedlings shown in **a**. The old and young mature leaves selected for further analyses were marked by red and white dotted frames, respectively. **d** Levels of Chl in the old and young leaves shown in **c**. **e** Steady-state levels of the indicated proteins in the old and young leaves analyzed in **c** were determined by immunoblot analysis using the indicated antibodies. **f** Representative images of 28-day-old WT, *bcm1-2*, and three independent *BCM2-OX/bcm1* transgenic lines overexpressing *BCM2* in the *bcm1-2* background grown under short-day normal light (120 μmol photon m$^{-2}$ s$^{-1}$) conditions. **g** qRT-PCR analysis of *BCM1* and *BCM2* transcripts in the seedlings shown in **f**. Expression levels are presented relative to those in WT seedlings. **h, i** Levels of Chl (**h**) and Mg-porphyrin (**i**) in 18-day-old WT, *bcm1-2* and *BCM2-OX/bcm1* seedlings grown under short-day normal light (120 μmol photon m$^{-2}$ s$^{-1}$) conditions. **j** Steady-state levels of the indicated proteins in seedlings analyzed in **h** were determined by immunoblot analysis using the indicated antibodies. In **a, c, f**, scale bars, 1 cm. In **b, g**, expression levels are presented relative to those in the VIGS-GFP/WT and WT seedlings, respectively. ND, not detected. In **b, d, g–i**, error bars represent SD of three biological replicates. Letters above histograms indicate significant differences determined by Tukey's HSD method (*P* < 0.05). In **e, j**, Ponceau S-stained RbcL was used as the loading control. Numbers below immunoblots represent normalized protein abundances relative to WT seedlings. Asterisks indicate nonspecific signals on the immunoblots.

deficiency or overproduction of BCM1 or BCM2 did not interrupt the up-regulation of *SAG12* and *SGR1* during DIS (Fig. 6e, f). These data suggest that both BCM isoforms act as conserved negative regulators of Chl breakdown.

To gain more insight into the consequences of Chl breakdown on the stability of the thylakoid membrane, we conducted electron microscopic analyses of chloroplast structure during DIS. Interestingly, the retarded rates of Chl degradation in the *BCM1/*

*2*-overexpressing plants were reflected in the retention of intact thylakoid membranes, whereas thylakoid membranes were almost completely degraded in *bcm1-3 bcm2-2* at 7 DDI (Fig. 6d). Since there is a strong correlation between disassembly of thylakoids and enlargement of plastoglobuli during senescence[52], plastoglobuli found in the *BCM1/2*-overexpressing plants were much smaller than in WT and *bcm1-3 bcm2-2* seedlings during dark incubation (Fig. 6d). These observations are consistent with

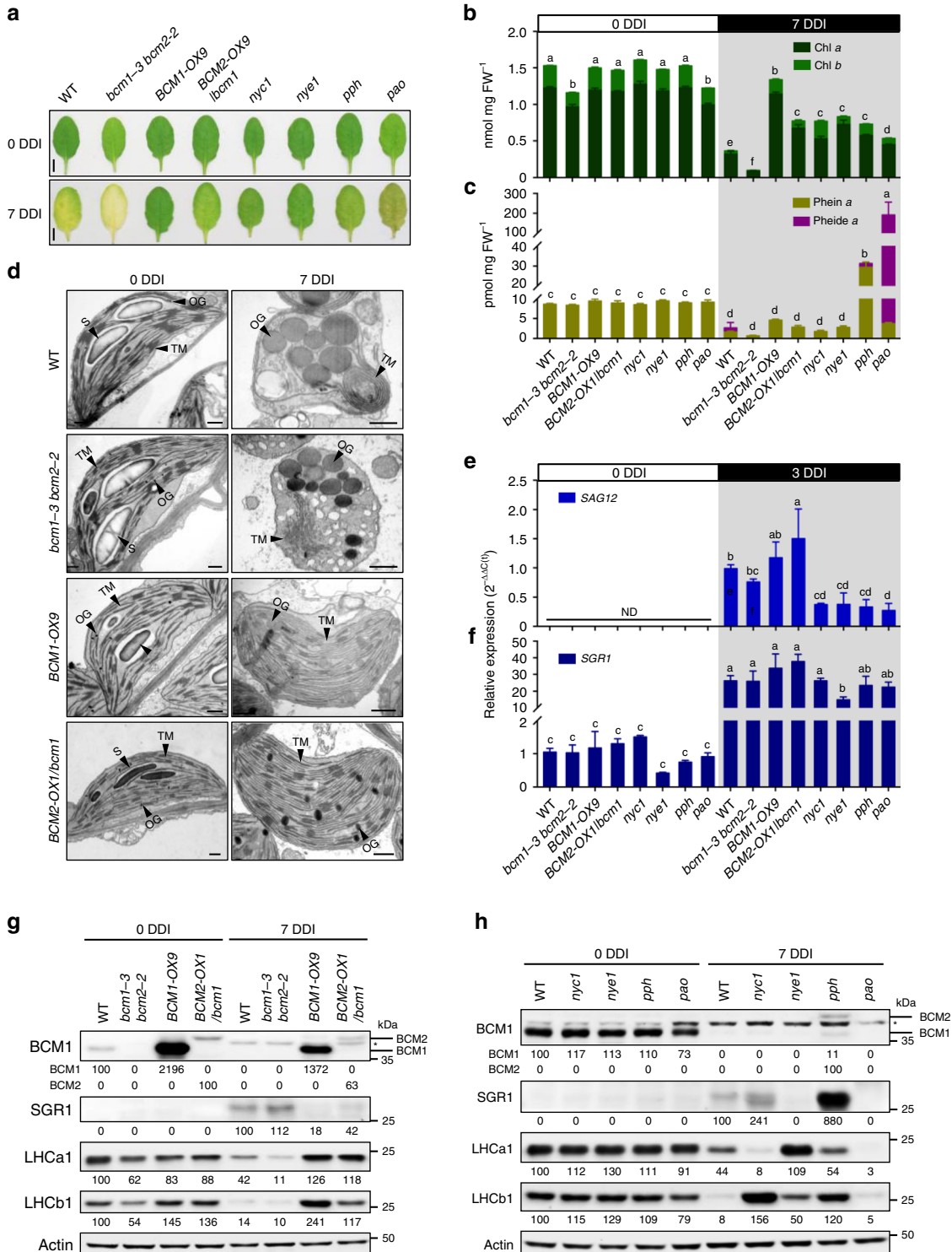

**Fig. 6 Both BCMs act as negative regulators of Chl catabolism. a** Representative images of detached leaves from 35-day-old *Arabidopsis* plants grown under short-day normal light (120 μmol photons m$^{-2}$ s$^{-1}$) conditions prior to (0 DDI) and after 7 days of dark incubation (7 DDI). Scale bars, 0.5 cm. **b, c** Levels of Chl (**b**) and Chl catabolites (**c**) in the detached leaves shown in **a** at 0 and 7 DDI. **d** Transmission electron micrographs showing chloroplast structures at 0 and 7 DDI. OG, osmiophilic plastoglobuli; S, starch granule; TM, thylakoid membrane. Scale bars, 0.5 μm. **e, f** qRT-PCR analysis of *SAG12* (**g**) and *SGR1* (**h**) transcripts in the detached leaves from 21-day-old *Arabidopsis* plants grown under short-day normal light (120 μmol photons m$^{-2}$ s$^{-1}$) conditions at 0 and 3 DDI. Expression levels of *SGR1* and *SAG12* are presented relative to those in the WT seedlings at 0 and 3 DDI, respectively. ND, not detected. **g, h** Steady-state levels of the indicated proteins in detached leaves from various *Arabidopsis* seedlings in **a** at 0 and 7 DDI were detected by immunoblotting using the indicated antibodies. Immunoblot analyses of Actin were used as a loading control. Numbers below the immunoblots represent normalized protein abundances relative to WT seedlings at 0 DDI. Asterisks indicate nonspecific signals on the immunoblots. In **b, c, e, f**, error bars represent SD of three biological replicates. Letters above histograms indicate significant differences determined by Tukey's HSD method ($P < 0.05$).

the hypothesis that Chl breakdown is a prerequisite for the degradation of PS-LHC complexes and disruption of thylakoid membranes[1,22,27].

Although *BCM1-OX* plants exhibited optimized Chl synthesis (Fig. 2b, c), the possibility that Chl breakdown might be directly inhibited by sustained synthesis of Chl in *BCM1-OX* plants was excluded based on the following findings: i. Continued Chl biosynthesis facilitated by ALA feeding to detached *Arabidopsis* leaves does not inhibit Chl breakdown (Supplementary Fig. 16). ii. Stability of TBS proteins interacting with BCMs, including GluTR, GUN4, and CHLM, was drastically reduced during DIS (Supplementary Fig. 17). iii. Although both BCM isoforms directly interact with GUN4 (Fig. 4a, b and Supplementary Fig. 13), overexpression of *GUN4* driven by the 35S promoter did not result in a stay-green phenotype during DIS (Supplementary Fig. 18). Thus, BCMs inhibit Chl breakdown independently of their function in Chl biosynthesis.

**BCMs delay Chl breakdown by destabilization of SGR1**. To dissect the molecular mechanism by which the two *Arabidopsis* BCM isoforms prevent Chl catabolism, the stay-green phenotypes of *BCM1/2-OX* plants were compared with those of Chl mutants defective in catabolic enzymes (Fig. 6). The *BCM1/2-OX* plants phenotypically mimic *nye1/sgr1*, a missense mutant of the predominant isoform of Mg-dechelatase[26,53], which retained Chl *a* and *b* equally well and exhibited stable accumulation of LHC proteins of both PSs during DIS (Fig. 6b, c, g, h). In contrast, *nyc1* only retained Chl *b*. Correspondingly, only its LHCII proteins were more stable than in WT at 7 DDI. Moreover, *pph*[28] and *pao*[29] accumulated dramatically higher levels of Phein *a* and Pheide *a*, respectively, under the same conditions (Fig. 6c, h).

Since neither disruption nor overexpression of *BCM1/2* has any impact on the expression of *CCG*s during dark incubation (Fig. 6f and Supplementary Fig. 14e), we next analyzed the steady-state levels of the two BCMs and SGR1 during dark incubation. Immunoblot analyses showed that BCM1 is degraded, whereas SGR1 accumulates during dark incubation (Fig. 6g). Although *BCM2* is suggested to be up-regulated during senescence, the BCM2 protein was barely detectable in immunoblot analyses of senescent leaves of WT plants, but was observed when *BCM2* was constitutively expressed (Figs. 5j and 6g), suggesting a very low rate of accumulation of BCM2 during DIS. Notably, we found that the content of SGR1 was diminished in *BCM1/2-OX* plants, but slightly increased in *bcm1-3 bcm2-2* compared to WT at 7 DDI (Fig. 6g). In contrast, *nyc1* and *pph* mutants showed much higher levels of SGR1 than WT during dark incubation (Fig. 6h). Surprisingly, both BCM1 and BCM2 were detectable in *pph* mutants at 7 DDI (Fig. 6h). However, the accumulation of SGR1 and both BCMs in *pph* mutants was not due to up-regulation of the corresponding transcripts (Supplementary Fig. 19). The retention of BCM1 and BCM2 in *pph* could in principle be explained with an inhibitory effect of the strong accumulation of Phein *a* on proteolysis of the two BCMs and SGR1 in *pph* during DIS. Based on these results, we conclude that overproduction of the two BCM paralogs destabilizes SGR1 during dark incubation.

The need for tight regulation of Mg-dechelatase is underlined by the finding that constitutively expressed *SGR1* induces Chl degradation during early leaf development[26,27]. A Y2H analysis of interactions between BCMs and Chl catabolic enzymes (CCEs) showed that both BCMs specifically interact with SGR1, and these findings were confirmed by BiFC and co-immunoprecipitation assays (Fig. 7a–c and Supplementary Fig. 20a, b). To further test the impact of the BCM isoforms on the stability of SGR1, we transiently overexpressed *SGR1* and *BCMs* in tobacco leaves. While overexpression of *SGR1* induced Chl breakdown, simultaneous

overexpression of *SGR1* together with *BCM1* or *BCM2* attenuated Chl breakdown, concomitantly with an overall reduction in SGR1 content (Fig. 7d–g and Supplementary Fig. 20c–f). To confirm these observations in *Arabidopsis*, we crossed *BCM1-OX* plants with *SGR1*-overexpressing (*SGR1-OX*) lines (Fig. 7h). Overexpression of *BCM1* indeed completely suppressed the negative effect of SGR1 overproduction on Chl levels and strongly attenuated accumulation of SGR1 (Fig. 7i–k). In conclusion, these results demonstrate that the two BCMs destabilize SGR1, and thus inhibit Chl breakdown.

**Discussion**

It is widely accepted that Chl biosynthesis is highly active during early leaf growth, whereas Chl is substantially degraded when plants enter the senescence phase[10,33]. However, the molecular mechanisms behind this fundamental phenomenon and the potential regulatory link between Chl biosynthesis and breakdown have not been discovered so far. In this study, we identified two evolutionarily conserved BCM proteins, which participate in the regulation of both anabolic and catabolic Chl pathways. Although Chl metabolism is transcriptionally regulated by light and phytohormone signaling[33,34], alterations in the levels of these BCMs in *Arabidopsis* do not interfere with the expression of either *CBG*s or *CCG*s (Fig. 6f and Supplementary Figs. 4a and 14e). This suggests that the two BCMs serve as concurrent post-translational regulators of Chl synthesis and catabolism.

Both BCM isoforms interact with the same proteins of Chl biosynthesis and catabolism—the MgCh-stimulating factor GUN4 and the dominant Mg-dechelatase isoform SGR1, respectively (Figs. 4 and 7 and Supplementary Figs. 13 and 20), suggesting functional preservation between the two BCMs. This notion is further supported by the findings. i. Disruption of BCM1 compromises Chl biosynthesis and results in a pale-green leaf phenotype in young seedlings, which can be rescued by ectopic expression of *BCM2* (Figs. 2 and 5). ii. Efficient inactivation of both *BCMs* leads to preferentially enhanced reduction in Chl content in old leaves rather than young leaves (Fig. 5a, c). iii. Constitutive expression of *BCM1* or *BCM2* retarded Chl degradation during senescence and conferred a cosmetic stay-green phenotype (Fig. 6 and Supplementary Fig. 14). Despite the similar functions of the two BCMs in Chl metabolism, their expression patterns are very different (Supplementary Fig. 9b, c). *BCM1* transcripts accumulate mainly in young green leaves, whereas *BCM2* is highly up-regulated during senescence. Accordingly, we propose a functional divergence between the two BCMs in the course of leaf development and senescence (Fig. 8). During early leaf development, when plants accumulate Chl to establish photosynthesis, BCM1 is the predominant isoform, acting to promote Chl biosynthesis by stimulating MgCh activity in the thylakoid membrane via interaction with GUN4 (Fig. 4c and Supplementary Fig. 6). Meanwhile, Chl catabolism is substantially suppressed in leaf tissue by a BCM1-dependent destabilization of SGR1, although the expression of *CCG*s is gradually up-regulated (Figs. 6g and 7f, j). During developmental transition from leaf maturation to senescence, activation of *SGR1* and *BCM2* is accompanied with inactivation of *BCM1* (Supplementary Fig. 9c). Both BCMs collectively restrict the accumulation of SGR1, thus decreasing the rate of Chl degradation. This notion is supported by the fact that deficiency of two BCMs leads to enhanced pale-green pigmentation of old leaves compared to those of *bcm1* mutant (Fig. 5a, c). Notably, BCM2 accumulated at very low level relative to BCM1 as BCM2 only can be immune-detectable in *BCM2-OX* seedlings (Fig. 5j). The reduced stability of BCM2 during DIS (Fig. 6g) allows for enhanced Chl degradation when

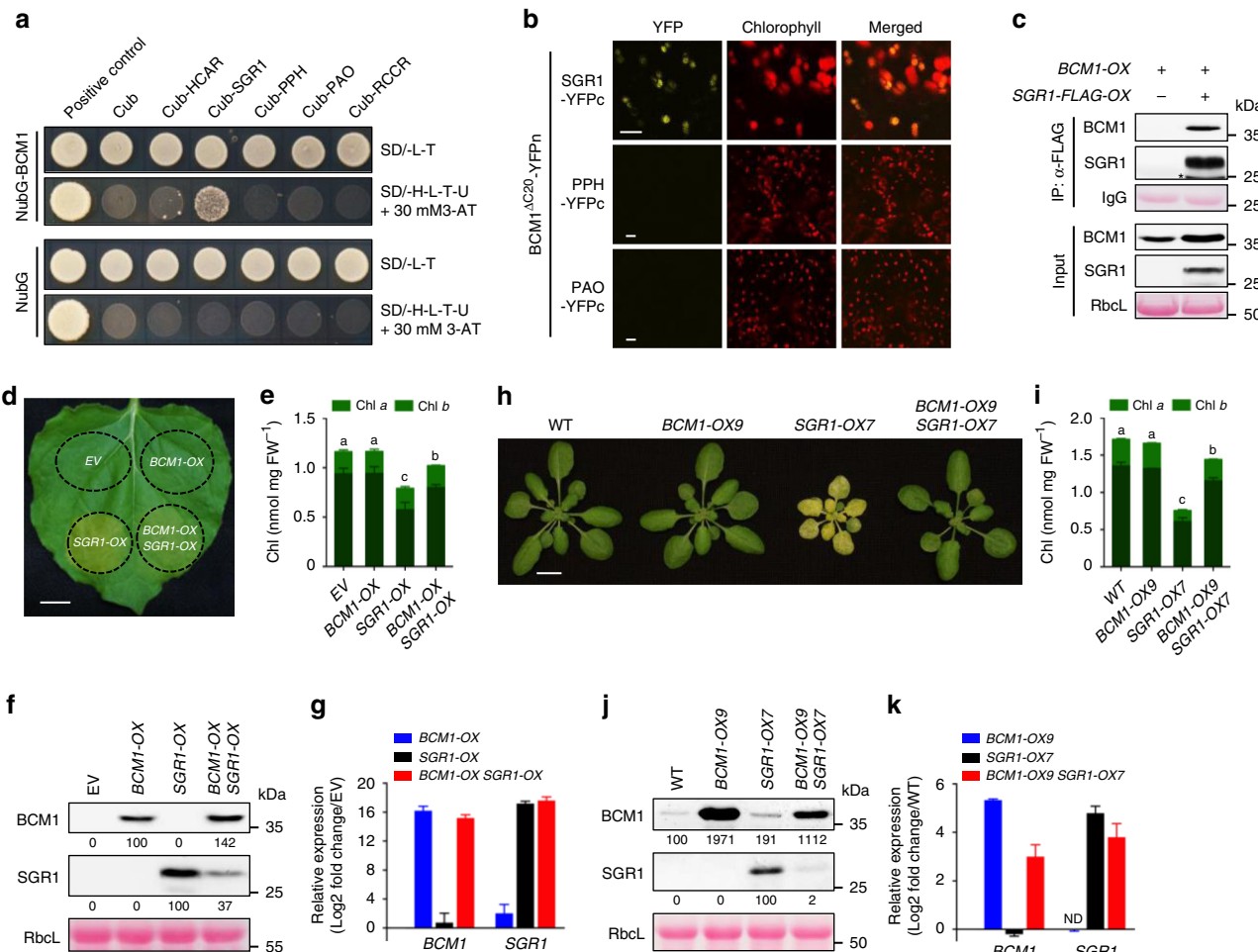

**Fig. 7 BCM1 physically interacts with and destabilizes SGR1. a** Y2H analysis of interactions between BCM1 and CCEs. The transformed yeast strains were analyzed on selective medium (SD/-L-T or SD/-H-L-T-U in the presence of 30 mM 3-AT). **b** BiFC assays confirm the specific interaction between BCM1 and SGR1. The combinations of BCM1$^{\Delta C20}$-nYFP with PPH/PAO-cYFP were used as negative controls. Scale bars, 10 μm. **c** Co-immunoprecipitation experiments demonstrate that BCM1 directly interacts with SGR1 in vivo. Anti-FLAG beads were used for immunoprecipitation. Samples of input and precipitated products were analyzed by immunoblot using anti-BCM1 and anti-SGR1 antibodies. **d** Representative image of a *N. benthamiana* leaf with zones overexpressing the empty vector (EV), *BCM1*, *SGR1*, and both *BCM1* and *SGR1* after 2 days of growth in the dark. The infiltrated leaf areas are indicated by circles. Scale bar, 1 cm. **e** Levels of Chl in the infiltrated leaf areas in **d**. **f** Steady-state levels of BCM1 and SGR1 in the infiltrated leaf areas in **d** were determined by immunoblotting using the indicated antibodies. **g** qRT-PCR analysis of *BCM1* and *SGR1* transcripts, confirming overexpression of these genes in the infiltrated leaf areas in **d**. Expression levels are presented relative to those in EV. **h** Representative image of 28-day-old WT, *BCM1-OX9*, *SGR1-OX7*, and *BCM1-OX9 SGR1-OX7* seedlings grown under short-day normal light (120 μmol photon m$^{-2}$ s$^{-1}$) conditions. Scale bar, 1 cm. **i** Levels of Chl in the seedlings shown in **h**. **j** Steady-state levels of BCM1 and SGR1 in the seedlings shown in **h** were determined by immunoblotting using the indicated antibodies. **k** qRT-PCR analysis of *BCM1* and *SGR1* transcripts, confirming overexpression of these genes in the seedlings shown in **h**. Expression levels are presented relative to those in WT seedlings. In **e**, **g**, **i**, **k**, error bars represent SD of three biological replicates. Letters above histograms indicate significant differences as determined by Tukey's HSD method ($P < 0.05$). In **c**, **f**, **j**, Ponceau S-stained membrane strips bearing RbcL or the light chain of IgG were used as loading controls. Numbers below the immunoblots represent normalized protein abundances in the examined genotypes relative to the control seedlings. Asterisk indicates nonspecific signals on the immunoblots. ND, not detected.

plant entered into the senescent stage. Thus, stimulation of Chl breakdown in the course of senescence seems to be a multifaceted process[1,33], which includes also inactivation of BCM2.

Multiple post-translational mechanisms in the chloroplast, which regulate the compartmental distribution of TBS enzymes and the organization of enzyme complexes, as well as proteolysis, thiol-based redox modification, and protein phosphorylation, contribute to a balanced output of TBS and strict light-dark control of Chl biosynthesis[10,39]. Our ongoing studies have identified many post-translational factors involved in regulating TBS, such as GluTR-BINDING PROTEIN (GBP)[54], the chaperone CHLOROPLAST SIGNAL RECOGNITION PARTICLE 43[55], and LHC-LIKE 3[56]. Intriguingly, only BCMs have a dual function

in Chl synthesis and catabolism. We propose that BCMs act as scaffold proteins to coordinate the Chl metabolic pathway[57]. Notably, their functions in Chl synthesis and catabolism apparently differ depending on the proteins with which they interact. Thus, interaction of BCM1 with GUN4 stimulates MgCh activity at the thylakoid membrane (Fig. 4c and Supplementary Fig. 6), whereas the interaction of either isoform with SGR1 reduces the stability of the enzyme (Figs. 6g and 7f, j and Supplementary Fig. 20e). How BCMs distinguish their target proteins and control two antagonistic metabolic pathways by two different mechanisms remains open. Three possibilities can be envisioned: (1) The two BCM isoforms preferentially interact with either GUN4 or SGR1. (2) Their interactions with GUN4 and SGR1 rely on differential

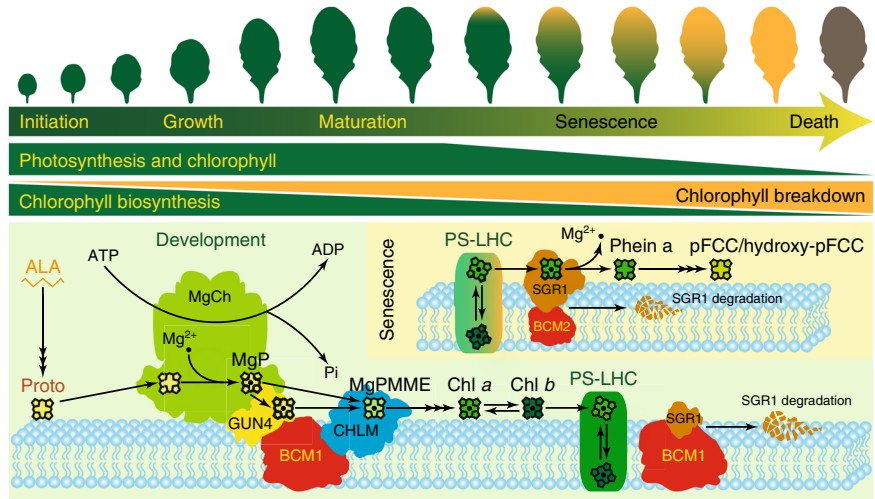

**Fig. 8 Model for the concurrent regulation of Chl biosynthesis and catabolism by BCMs.** During early leaf development (encompassing leaf initiation, growth, and maturation), optimized Chl biosynthesis ensures efficient photosynthesis, while Chl breakdown is largely suppressed. BCM1 promotes Chl biosynthesis by stimulating MgCh through GUN4 action, and in addition interacts with and destabilizes SGR1 to prevent Chl degradation. Upon the onset of leaf senescence, although BCM1 is greatly suppressed, *BCM2* and *SGR1* are both up-regulated. BCM2 is therefore able to contribute to the inhibition of Chl breakdown. However, the strong accumulation of SGR1 eventually allows plants to initiate Chl breakdown.

interaction domains. (3) Unknown post-translational modifications of BCMs influence their binding affinity for their target proteins. These hypotheses await future investigation.

Although structural and mechanistic studies on MgCh have been undertaken for decades, the precise mechanism underlying the insertion of Mg into Proto remains unclear[9,10,58]. The most widely accepted model is that the MgCh complex is transiently formed as a holoenzyme by the initial assembly of two interacting hexameric rings consisting of CHLD and CHLI, which then interact with the catalytic subunit CHLH. Driven by CHLI-mediated ATP hydrolysis, CHLH catalyzes the insertion of Mg into Proto before the holoenzyme is disassembled for the next catalytic cycle[11,12,59]. GUN4 serves as an accessory protein to stimulate MgCh activity by promoting its interaction with CHLH and binding of Proto and MgP[13–15,59,60]. Our findings introduce BCM1 as another important auxiliary factor for MgCh. *bcm1* mutants phenotypically resemble MgCh mutants[13,15,46] (e.g., *gun5-1* and *gun4-1*) that exhibit reduced rates of ALA synthesis, and lower levels of Mg-porphyrins and Chl, but not of heme (Fig. 2b–g). More importantly, the similarity between the Chl biosynthesis phenotypes observed in *gun4-1* and *bcm1-3 gun4-1* mutants leads to the conclusion that BCM1 and GUN4 function in parallel to promote MgCh activity in planta (Fig. 3a–e). In accordance with these findings, BCM1 physically interacts with GUN4 but not with CHLH (Fig. 4a, b), and exclusively promoted MgCh activity in the presence of GUN4 in vitro (Fig. 4c and Supplementary Fig. 6). We therefore propose that BCM1 facilitates Mg chelation by interacting with GUN4.

Although the results presented here shed light on the positive role of BCM1 in the MgCh step, it remains unclear how and where the MgCh complex is assembled in the chloroplast. Due to the increased hydrophobicity of TBS intermediates, Chl biosynthesis initiated by MgCh preferentially takes place at the chloroplast membranes[3,61]. It has been suggested that GUN4-porphyrin complexes promote the association of CHLH with chloroplast membranes, thus optimizing the channeling of Chl intermediates within the pathway[60,62]. However, it is still not clear how the GUN4-MgCh complex is associated with the membrane. Here, we propose that the integral membrane protein BCM1 acts as a scaffold for the transient membrane contact of the active GUN4-MgCh complex (Fig. 8). As CHLM has been reported to assemble

with MgCh to facilitate the trafficking of MgP from CHLH to CHLM[44], it is suggested that the observed interaction between BCM1 and CHLM (Fig. 4a, b) promotes the organization of a MgCh-GUN4-CHLM enzyme complex.

While many post-translational control mechanisms have been described in the Chl biosynthetic pathway[10,39], post-translational control of Chl catabolism, including the activity, stability, and suborganellar localization of SGR1, awaits further study[22,33]. Genetic studies have shown that overproduction of SGR isoforms, such as SGR1 and SGR-LIKE (SGRL), rather than PPH and PAO, could induce Chl breakdown in young green leaves (Fig. 7d, h and Supplementary Fig. 20c)[26,27,63], highlighting the tight control of the steady-state level of SGR. We found the *BCM1/2-OX* plants showed a *nye1*-like cosmetic stay-green phenotype (Fig. 6 and Supplementary Fig. 14). Both BCM isoforms physically interact with SGR1 in vitro and in vivo (Fig. 7a–c and Supplementary Fig. 20a, b). Intriguingly, the SGR1 content was reduced in *BCM1/2-OX* seedlings (Figs. 6g and 7f, j and Supplementary Fig. 20e), whereas *nyc* and *pph* mutants accumulated more SGR1 than WT during senescence (Fig. 6h). These results are interpreted to mean that both BCM variants cooperatively control the precise level of SGR1 from young leaf development up to senescence and direct excess SGR1 into a proteolysis pathway mediated by an unknown protease (Fig. 8). Identification of the protease involved in the quantitative control of SGR during leaf development remains a challenge for future research.

Mutations in SGR1/NYE1 orthologs cause a stay-green phenotype in many land plants, such as *Arabidopsis*[26], rice[64], pea[65], tomato[66], and *Brassica napus*[67]. However, the SGR homolog in *Chlamydomonas reinhardtii* is suggested to be required for PSII formation rather than Chl degradation[68], suggesting that SGR has functionally diverged during the evolution of oxygen-dependent photosynthetic organisms. BCM orthologs are found in angiosperms, gymnosperms, and bryophytes, but not in cyanobacteria or algae (Supplementary Fig. 8). We propose an evolutionarily conserved function for BCMs in determining Chl levels during leaf development, since knockout mutants for BCM orthologs in *Arabidopsis*, soybean, and rice display a pale-green leaf phenotype (Fig. 2)[42]. It was recently reported that the *BCM* orthologs have been selected during the domestication of soybean, rice and tomato[42]. Therefore, we infer that the emergence of BCMs might

be correlated with the functional divergence of SGR during the evolution of land plants.

The tight regulation of Chl metabolism is not only essential for the precise adjustment of photosynthetic capacity but is also implicated in other important biological processes, such as plastid-to-nucleus retrograde signaling[13,46,69], abscisic acid (ABA) signaling[70], and RNA editing[71]. While both CHLH and GUN4 are suggested to contribute to plastid retrograde signaling[13,46], whether both BCMs also participate in plastid retrograde signaling via their interaction with GUN4 remains to be addressed. It has been shown that CHLH serves as an ABA receptor to perceive the ABA signal and trigger the downstream signaling cascade[70]. Interestingly, BCM's orthologs in Arabidopsis, soybean, and rice were shown to regulate seed dormancy via interaction with ABA-synthesizing enzymes and in turn modulate ABA synthesis[42]. Because ABA signaling can efficiently induce Chl breakdown, the previously observed increased ABA level in the BCM orthologs-overexpressing plants[42] was not correlated with the cosmetic stay-green phenotype and WT-like expression of CCGs observed in BCM1/2-OX plants during DIS (Fig. 6 and Supplementary Fig. 14). Nevertheless, the molecular function of BCMs in ABA synthesis remains to be investigated[42]. Further studies will precisely define the multifaceted molecular functions of BCMs and might unravel potential strategies to improve quality traits of crop plants.

## Methods

**Plant materials and growth conditions**. The *Arabidopsis thaliana* ecotype Columbia-0 (Col-0) was used as WT. Mutant lines used in this study are listed in Supplementary Table 1. Double mutants were obtained by crossing mutant lines, and all genotypes were confirmed by PCR-based genotyping or immunoblot analyses. Primers used for genotyping are listed in Supplementary Table 2. After a 2-day stratification in darkness at 4 °C, *Arabidopsis* seeds were sown in soil, and germinating young seedlings were grown under standard short-day conditions (10 h light/14 h dark, 120 μmol photons $m^{-2} s^{-1}$, 23 °C, and 70% relative humidity). For analyses of pigment, gene expression and protein, each sample consisting of 4–6 individual 18-day-old seedlings was harvested. For VIGS assay, the first two true leaves of 14-day-old *Arabidopsis* plants grown under long-day conditions (16 h light/8 h dark, 120 μmol photons $m^{-2} s^{-1}$, 23 °C, and 50% relative humidity) were infiltrated with the *A. tumefaciens* strain GV2260 harboring the respective constructs[51]. For DIS analyses, the detached 4th to 6th rosette leaves of 35-day-old plants grown under standard growth conditions were placed on wet paper towels and incubated in darkness for the indicated times. For protein interaction and protein stability analyses, 3-week-old *Nicotiana benthamiana* plants grown under long-day conditions were used to transiently express proteins of interest.

**Plasmid construction and plant transformation**. All primers used to generate plasmid constructions are listed in Supplementary Table 3. For plant transformation, the full-length coding sequences (CDSs) of *BCM1*, *BCM2*, *GUN4*, and *SGR1-FLAG* (containing a carboxyl terminal FLAG tag) were cloned into the *pGL1* binary vector, in which gene expression is driven by the constitutive cauliflower mosaic virus 35S promoter[55]. Transgenic *Arabidopsis* lines constitutively expressing *BCM1*, *BCM2*, *GUN4*, and *SGR1-FLAG* were generated by transforming Col-0 or *bcm1-2* plants with the *A. tumefaciens* strain GV2260 harboring the respective constructs. The $T_1$ generation of transgenic lines was selected based on their BASTA herbicide resistance. After segregation of $T_2$ plants, homozygous $T_3$ progenies were used for further studies. To analyze the influence of BCM1 on the stability of SGR1, the lower epidermal cells of *N. benthamiana* plants were infiltrated with the *A. tumefaciens* strain GV2260 carrying the respective constructs and incubated in darkness for 2 days.

For subcellular localization analysis, the full-length CDS or cTP of *BCM1* were cloned into the *pUC19-YFP* vector to yield a fusion protein with YFP at the C terminus. For generation of the anti-BCM1 antibody, the nucleotide sequence encoding BCM155–253 was cloned into the expression vector pET28a (Novagen, Germany). For expression and purification of recombinant BCM1 in the *Saccharomyces cerevisiae*, the nucleotide sequence encoding mature BCM1 was cloned into pDR296 expression vector. For VIGS assay, the nucleotide sequence encoding BCM2117–334 was cloned into *pTRV2* vector. For BiFC assays, the full-length CDSs of *GluTR*, *GUN4*, *CHLH*, *CHLM*, *SGR1*, *PPH*, and *PAO* were cloned either into the *pDEST-GW-VYNE* (to express fusion proteins containing the N-terminal half of YFP, nYFP) or *pDEST-GW-VYCE* (to fuse each gene to the C-terminal part of YFP, cYFP). To maintain the split YFP sequence stably fused to the C terminus of BCM1, the nucleotide sequence encoding BCM1$^{\Delta C20}$ (lacking the last 20 amino acids of the mature protein) was cloned into the BiFC vectors. For split ubiquitin-based Y2H assays, the nucleotide sequences encoding mature BCM1

and BCM2 were cloned into the *pXNgate* prey vector containing NubG (Dualsystems Biotech, Schlieren, Switzerland). The nucleotide sequences encoding mature GluTR, GBP, GSAT, protoporphyrinogen oxidase, GUN4, CHLH, CHLD, CHLI, CHLM, CHL27, the B isoform of POR, CHLG, CAO, HCAR, SGR1, PPH, PAO, and RCCR were cloned into the *pDHB1* vector (Dualsystems Biotech, Schlieren, Switzerland) for the interaction study with *pXNgate* fusion vectors. For the yeast complementation assay, sequences encoding the mature BCM1, BCM2, RCE1, and STE24 proteins were cloned into the yeast expression vector pJR1138[50].

**RNA extraction and qRT-PCR**. Total RNA was isolated from leaf materials using the citric acid method[55]. Aliquots (2 μg) of DNase-treated RNA were used to synthesize first-strand cDNA with RevertAid reverse transcriptase (Thermo Fisher Scientific, Waltham, MA, USA) and oligo dT(18) primer. Gene expression was determined by qRT-PCR using 2× qPCR mastermix (Bimake, Houston, TX, USA) and the CFX96-C1000 96-well plate thermocycler (Bio-Rad, Hercules, CA, USA). Transcription levels of specific genes were normalized to that of *SAND* (*At*2g28390) and *ACTIN2* (*At*3g18780) and calculated with the Bio-Rad CFX-manager software (1.6) using the ΔΔC(t) method. For qRT-PCR analyses in *N. benthamiana* plants, *α-TUBULIN* (AJ421411) was used as the reference gene. In addition, semiquantitative reverse transcription-PCR (RT-PCR) was used to confirm mutation of genes of interest in the T-DNA insertion mutants. In such cases, *UBIQUITIN10* (*At*4g05320) was used as the internal control. Primers for RT-PCR and qRT-PCR are listed in Supplementary Table 4.

**HPLC analysis of tetrapyrroles**. Pigments (Chls and carotenoids), and heme and Chl metabolites (precursors and catabolites) were analyzed by high-performance liquid chromatography (HPLC)[72,73]. Rosette leaves (30–50 μg) were harvested from the indicated ecotypes, weighted to determine their fresh weight (FW), and ground in liquid nitrogen. Chl, carotenoid, and Chl metabolites were extracted from frozen leaf powder using 500 μL of ice-cold acetone:0.2 M $NH_4OH$ (9:1, v/v) and centrifuged at $16,000 \times g$ for 20 min at 4 °C. Non-covalently bound heme was extracted from the pellet remaining after pigment extraction using 200 μL of AHD buffer (acetone:hydrochloric acid:dimethylsulfoxide, 10:0.5:2, v/v/v), and centrifuged at $16,000 \times g$ for 20 min at room temperature (RT). The supernatants were analyzed by HPLC using the Agilent 1100 or 1290 HPLC system equipped with a diode array and fluorescence detectors (Agilent Technologies, Santa Clara, CA, USA). Identification and quantification of pigments, heme, and Chl metabolites were carried out with the aid of authentic standards.

**Determination of ALA synthesis capacity**. Whole rosette seedlings were excised from 18-day-old plants, weighted to determine FW, and incubated in 5 mL of reaction buffer (50 mM Tris-HCl, pH 7.2, and 40 mM levulinic acid) for 3 h under standard growth conditions. Tissues were then ground in liquid nitrogen and resuspended in 500 μL of 20 mM potassium phosphate buffer (pH 7.2). After centrifugation at $12,000 \times g$ for 5 min at 4 °C, 400 μL of the supernatant was mixed with 100 μL of ethyl acetoacetate (Sigma-Aldrich, St. Louis, MO, USA) and boiled for 10 min. After cooling on ice, 500 μL of Ehrlich's reagent (373 mL of acetic acid, 90 mL of 70% [v/v] perchloric acid, 1.55 g of $HgCl_2$, 9.10 g of 4-dimethylamino benzaldehyde, and 500 mL of $ddH_2O$) was added, and the mixture was centrifuged at $12,000 \times g$ for 5 min at 4 °C. The absorption of the ALA pyrrole was measured at 525, 553, and 600 nm. The ALA content was calculated using a calibration curve generated with authentic ALA (Sigma-Aldrich, St. Louis, MO, USA) and was normalized to the FW and incubation time.

**CHLM enzymatic assay**. Rosette leaves were harvested from 21-day-old WT and three *bcm1* mutant plants, weighted to determine FW, ground in liquid nitrogen, and resuspended in reaction buffer (0.3 M sorbitol, 20 mM Tricine-KOH, pH 8.4, 2.5 mM EDTA, 5 mM $MgCl_2$, and cOmplete protease inhibitor cocktail). The reaction was initiated by mixing the extract with an equal volume of assay buffer supplemented with 500 μM *S*-adenosylmethionine and 10 μM MgP, and incubated in darkness for 15 min at 30 °C. The reaction was stopped by the addition of acetone:0.2 M $NH_4OH$ (9:1, v/v). After centrifugation at $16,000 \times g$ for 20 min at 4 °C, the level of MgPMME produced was determined by HPLC. The CHLM enzymatic activity was calculated by normalization to the FW of sample and incubation time.

**In vitro MgCh enzymatic assay**. The in vitro MgCh assay was performed as described previously[15,47]. The recombinant MgCH subunits from rice and GUN4 from *Arabidopsis* were expressed and purified from *Escherichia coli* BL21 (DE3) or Rosetta (DE3) cells. The recombinant His-tagged BCM1 was expressed driven by pDR296-BCM1 in the *Saccharomyces cerevisiae* strain L40ccU A. As a control, the empty vector pDR296 was transformed into the L40ccU A strain. The total yeast membranes isolated from the yeast cells grown in SD/-Trp liquid medium were resuspended in phosphate buffer. The recombinant His-BCM1 was purified from the isolated yeast membranes, which were solubilized with 1% [w/v] β-DM. A typical MgCh assay (overall 150 μL) contained 2.5 μM CHLH, 1 μM CHLD, 1 μM CHLI, 2.5 μM GUN4, 30 μg protein of yeast membranes, 1 μM His-BCM1 containing 0.15 mM β-DM, and 1 μM GST in MgCh assay buffer (50 mM Tricine-NaOH, pH 8.0, 15 mM $MgCl_2$, 2 mM dithiothreitol [DTT], 1 mM ATP, and 10 μM

Proto). The CHLH, GUN4, yeast membranes, His-BCM1, GST, and Proto in MgCH assay buffer (overall 100 μL) were pre-incubated in a reaction tube in darkness for 30 min on ice, while CHLD and CHLI were separately pre-incubated in 50 μL of MgCH assay buffer. The equal amount of β-DM was supplemented in each reaction when BCM1 was present in the MgCh assay. The reaction was started by mixing the components, and the reaction mixtures were incubated in darkness for 45 min at 30 °C. A 100-μL aliquot of the reaction solution was then mixed with 400 μL of acetone:0.2 M NH$_4$OH (9:1, v/v). After centrifugation at 16,000 × g for 20 min at 4 °C, the supernatant was analyzed by HPLC to measure the level of MgP produced. A 30-μL sample of reaction solution, supplemented with 10 μL of standard Laemmli buffer, was boiled and loaded on 12% sodium dodecyl sulfate–polyacrylamide gel electrophoresis (SDS-PAGE) to determine the protein levels after the reaction. The MgCh activity was normalized to incubation time.

**Subcellular localization analysis**. To determine the subcellular localization of the BCM1 protein, we transformed the plasmids (pUC-YFP [a negative control], pUC-BCM1-YFP, and pUC-cTP$_{BCM1}$-YFP) into Arabidopsis mesophyll protoplasts as described previously[74]. After 12 h of incubation in darkness, the subcellular localization of BCM1-YFP was analysed using excitation/emission filters for YFP fluorescence (Ex/Em, 514/530–555 nm) and Chl fluorescence (Ex/Em, 514/600–700 nm) with a confocal laser-scanning microscope Leica TCS SP2 (Leica Microsystems, Wetzlar, Germany).

**Chloroplast isolation and fractionation**. To isolate intact chloroplasts, 4-week-old Arabidopsis plants grown under standard conditions were harvested and homogenized in isolation buffer (0.45 M sorbitol, 20 mM Tricine-KOH, pH 8.4, 10 mM EDTA, 10 mM NaHCO$_3$, and 0.1% [w/v] bovine serum albumin) using a blender equipped with sharp razor blades. The homogenate was filtered through two layers of Miracloth (Calbiochem, San Diego, CA, USA) and centrifuged at 1000 × g for 5 min at 4 °C. The pellets were gently resuspended in resuspension buffer (0.3 M sorbitol, 20 mM Tricine-KOH, pH 8.0, 5 mM MgCl$_2$ and 2.5 mM EDTA), and loaded onto Percoll step gradients (40% [v/v] and 80% [v/v]). After centrifugation at 6500 × g for 15 min at 4 °C, chloroplasts were collected from the interface between the Percoll suspensions, washed twice with resuspension buffer, and finally resuspended in 1 mL of resuspension buffer for further analyses.

For chloroplast fractionation, freshly isolated chloroplasts were lysed in hypertonic TE buffer (10 mM Tris-HCl, pH 8.0 and 2 mM EDTA), adjusted to a Chl concentration of 2 μg μL$^{-1}$ on ice for 10 min, and loaded on a three-step sucrose gradient, consisting of 2 mL of 1.2 M sucrose, 3 mL of 1 M sucrose, and 3 mL of 0.46 M sucrose. After ultracentrifugation at 200,000 × g for 1 h at 4 °C, fractions of stroma, envelope membrane, and thylakoid membrane were collected from the supernatant, the 0.46 M/1 M sucrose interface, and the pellet, respectively. Isolated chloroplasts and chloroplast fractions were either used immediately or frozen in liquid nitrogen and stored at −80 °C in aliquots.

**Isolation and subfractionation of thylakoid membranes**. Total thylakoid membranes were isolated by fractionation of chloroplasts or isolated from rosette leaves[75]. To prepare grana core-, grana margin-, and stromal lamellae-enriched thylakoid membranes, freshly isolated thylakoids were diluted in resuspension buffer (15 mM Tricine, pH 7.8, 100 mM sorbitol, 10 mM NaCl, and 5 mM MgCl$_2$) to a final Chl concentration of 0.5 μg μL$^{-1}$, supplemented with 0.8% (w/v) digitonin (Calbiochem, San Diego, CA, USA), and solubilized in darkness for 5 min at RT. The reactions were stopped by the addition of eight volumes of resuspension buffer. After centrifugation at 1000 × g for 3 min at 4 °C, the supernatant was transferred to a new tube and centrifuged at 10,000 × g for 30 min at 4 °C to pellet grana core-enriched thylakoids. To collect grana margin-enriched thylakoids, the supernatant was then ultracentrifuged at 40,000 × g for 30 min at 4 °C. To collect stromal lamellae-enriched thylakoids, the supernatant was ultracentrifuged at 145,000 × g for 1 h at 4 °C.

**Clear Native-PAGE analysis**. Freshly isolated thylakoid membranes (8 μg of Chl) were resuspended in 25BTH20G (25 mM BisTris-HCl, pH 7.0 and 20% [v/v] glycerol), supplemented with 1% (w/v) β-DM and solubilized in darkness for 10 min on ice[75]. The β-DM-insoluble materials were pelleted by centrifugation at 16,000 × g for 10 min at 4 °C. The supernatant was supplemented with 0.3% (w/v) sodium deoxycholate (DOC). For electrophoresis, 50 mM BisTris-HCl (pH 7.0) was used as the anode buffer, and the cathode buffer (50 mM Tricine, 15 mM BisTris-HCl, pH 7.0) was supplemented with 0.02% (w/v) DOC and 0.05% (w/v) β-DM. Electrophoresis was conducted on ice, starting with an applied voltage of 50 V for 30 min. A gradual increase in the voltage was then applied to maintain constant current until the samples reached the end of the gel. For the second dimensional SDS-PAGE, excised Clear Native-PAGE lanes were immersed in SDS sample buffer (50 mM Tris-HCl, pH 6.8, 2% [w/v] SDS, 10% [v/v] glycerol, 0.002% [w/v] bromophenol blue, and 50 mM DTT) for 1 h at RT, and then loaded onto 11% SDS-urea-PA gels containing 6 M urea to dissociate the individual complexes and separate their components. After electrophoresis, the SDS-polyacrylamide gels were stained with Coomassie Brilliant Blue R250.

**Protein extraction and immunoblot analysis**. Whole rosette seedlings or the 4th–6th rosette leaves were harvested from 4 to 6 individual plants for each genotype, ground in liquid nitrogen, and supplemented with 500-800 μL of PEB buffer (2% [w/v] SDS, 56 mM NaCO$_3$, 12% [w/v] sucrose, and 2 mM EDTA, pH 8.0). The mixture was thawed at RT and heated at 70 °C for 20 min. The isolated chloroplasts and thylakoid membranes were directly resuspended in PEB buffer and incubated at 70 °C for 20 min. After centrifugation at 14,000 × g for 10 min at RT, the supernatants were transferred to a new reaction tube. The protein concentration was determined using the Pierce BCA Protein Assay Kit (Thermo Fisher Scientific, Waltham, MA, USA) according to the manufacturer's instructions. All samples were then diluted to the same protein concentration in PEB buffer supplemented with 56 mM DTT as a reducing agent, and incubated at 70 °C for 5 min. Twelve micrograms of total leaf proteins or 4 μg of thylakoid proteins was loaded per lane and fractionated on 12% SDS-PA gels or 12% SDS-urea-PA gels containing 6 M urea.

After electrophoresis, proteins were transferred to nitrocellulose membranes (GE Healthcare, Chicago, IL, USA) and probed with specific antibodies. Antibodies against BCM1 (dilution: 1:500), GluTR (dilution: 1:1000), GSAT (dilution: 1:2000), GUN4 (dilution: 1:2000), and CHLM (dilution: 1:500) were generated in the lab[15,55], those for CHL27 (AS06122, dilution: 1:1000), D1 (AS05084, dilution: 1:5000), PsaL (AS06108, dilution: 1:2500), Cyt b$_6$ (AS184169, dilution: 1:2500), CF1β (AS05085, dilution: 1:5000), Tic40 (AS10709, dilution: 1:2500), LHCa1 (AS01005, dilution: 1:2500), and LHCb1 (AS09522, dilution: 1:2500) were purchased from Agrisera (Vännäs, Sweden), and the anti-FLAG antibody (B23101, dilution: 1:1000) was obtained from Sigma-Aldrich (St. Louis, MO, USA). Antibodies against CHLH (dilution: 1:1000), CHLI (dilution: 1:5000), and SGR1 (dilution: 1:500) were kindly provided by Dr. Da-Peng Zhang (Tsinghua University, China), Dr. Meizhong Luo (Huazhong Agricultural University, China), and Dr. Ayumi Tanaka (Hokkaido University, Japan), respectively. Immunoblotting signals were induced by addition of the SuperSignal West Pico Chemiluminescent Substrate (Bio-Rad, Hercules, CA, USA) and detected with a CCD camera (Intas Biopharmaceuticals, Ahmedabad, India).

**Generation of a polyclonal antibody against BCM1**. The pET28a- BCM155–253 construct was transformed into Escherichia coli BL21 (DE3) cells. Expression of the recombinant His-BCM55–253 protein was induced by the addition of 0.4 mM isopropyl β-D-1-thiogalactopyranoside for 3 h at 37 °C. The overexpressed proteins were solubilized in denaturing buffer (200 mM NaCl, 8 M urea, and 50 mM Tris-HCl, pH 8.0) and purified under denaturing conditions using Ni-NTA agarose (Thermo Fisher Scientific, Waltham, MA, USA) according to the manufacturer's user guide. The purified proteins were concentrated and buffer exchanged by passage through Amicon Ultra-4 Centrifugal Filter Units (MWCO 3 kDa, Merck-Millipore, Burlington, VT, USA), aliquoted, and stored at −80 °C in phosphate-buffered saline buffer containing 0.8 M urea and 5% (v/v) glycerol. A polyclonal antibody against BCM1 was raised in rabbit by injection of the purified His-BCM55–253 (Biogenes, Berlin, Germany). Antisera were affinity purified by incubation with the same antigen coupled to nitrocellulose membranes.

**Transmission electron microscopy**. The 4th–6th rosette leaves were excised from 35-day-old Arabidopsis plants and incubated in darkness for 0 and 7 days. Leaf pieces of about 1.5 × 1.5 mm were then cut with a sharp razor blade and immediately immersed in fixation buffer (0.1 M sodium phosphate buffer, pH 7.4, 2.5% [v/v] glutaraldehyde, and 4% [v/v] formaldehyde) at RT. A mild vacuum (about 20 mbar) was applied until the leaf pieces sank, the fixation buffer was then replaced and the samples were fixed overnight at 4 °C. After three 10-min washes in sodium phosphate buffer (pH 7.4), the samples were osmicated with 1% (w/v) osmium tetroxide and 1.5% (w/v) potassium ferricyanide in 0.1 M sodium phosphate buffer (pH 7.4) for 60 min at 4 °C. The samples were rinsed three times for 10 min each in ddH$_2$O, and incubated in 1% (w/v) uranyl acetate at 4 °C overnight. After three 10-min washes in ddH$_2$O, the samples were embedded in 1% (w/v) Difco™ Agar noble (Becton, Dickinson and Company, Sparks, MD, USA), dehydrated using increasing concentrations of ethanol, and embedded in glycid ether 100 (formerly Epon 812; Serva, Heidelberg, Germany) with propylene oxide as intermediate solvent following standard procedures. Polymerization was carried out for 40–48 h at 65 °C. Ultrathin sections (~60 nm) were cut with a diamond knife (type ultra 45°; Diatome, Biel, Suisse) on an EM UC7 ultramicrotome (Leica Microsystems, Wetzlar, Germany) and mounted on single-slot Pioloform-coated copper grids (Plano, Wetzlar, Germany). The sections were stained with uranyl acetate and lead citrate and viewed with a JEM-2100 transmission electron microscope (JEOL, Tokyo, Japan) operated at 80 kV. Micrographs were taken using a 4080 × 4080 or 1350 × 1040 pixels charge-coupled device camera (UltraScan 4000 or Erlangshen ES500W, respectively, Gatan, Pleasanton, CA, USA) and Gatan DigitalMicrograph software (version1.85.1535).

**BiFC assay**. The appropriate constructs were transiently transformed into the lower epidermal cells of N. benthamiana leaves using A. tumefaciens strain GV226055. After infiltration, the tobacco plants were grown in darkness for 3 days and the leaf segments from infiltrated areas were then examined with a confocal laser-scanning microscope Leica TCS SP2 (Leica Microsystems, Wetzlar,

Germany). YFP signals were detected at Ex/Em 514/530–555 nm, while Chl fluorescence was visualized at Ex/Em 514/600–700 nm.

**Split ubiquitin-based Y2H assay.** The Y2H assays were performed as described in the manufacturer's user guide (Dualsystems Biotech, Schlieren, Switzerland). The empty *pXNgate* and *pDHB1* vectors served as negative controls. The combination of NubG-GluTR and Cub-GBP was used as a positive control. The resulting *pXNgate* and *pDHB1* fusion vectors were transformed into the *Saccharomyces cerevisiae* strains L40ccU α and L40ccU A, respectively, using the standard lithium acetate transformation protocol. After mating, yeast cells containing both *pXNgate* and *pDHB1* vectors were selected on SD/-Leu-Trp agar plates, and plated onto SD/-His-Leu-Trp-Ura agar supplemented with 30 mM 3-amino-1,2,4-triazole (Sigma-Aldrich, St. Louis, MO, USA) to test for positive interactions.

**Co-immunoprecipitation.** The *A. tumefaciens* strain GV2260 harboring the indicated constructs for transient transformation of *N. benthamiana* leaves were obtained as described for the BiFC assay. After a 2-day dark incubation, intact chloroplasts were isolated from infiltrated leaves and solubilized in protein extraction buffer (20 mM HEPES-KOH, pH 8.0, 1.1% Triton X-100 [v/v], 0.2% [v/v] NP-40, 200 mM NaCl, and cOmplete protease inhibitor cocktail) for 15 min at 4 °C. After centrifugation at $20,000 \times g$ for 20 min at 4 °C, the supernatant (1 mg) was mixed with 10 μL of Anti-FLAG Affinity beads (Bimake, Houston, TX, USA). After 2 h of incubation at 4 °C, the beads were washed three times with wash buffer (20 mM HEPES-KOH, pH 8.0, 0.1% Triton X-100 [v/v], 0.2% [v/v] NP-40, 200 mM NaCl, and cOmplete protease inhibitor cocktail). The SGR1-FLAG and its interacting proteins were then eluted from the beads with standard Laemmli buffer and subjected to SDS-PAGE and immunoblot analyses.

**Pheromone diffusion (halo) assay.** To perform the halo assay, the indicated constructs were transformed into the WT (JRY6958) and *rce1Δ ste24Δ* mutant (JRY6959) *MATa* yeast strains using a standard lithium acetate transformation protocol[50]. Yeast peptone-dextrose plates containing 0.01% (v/v) Triton X-100 were spread with a lawn of the *MATα sst2* cells (JRY3443), and 5 μL of the *MATa* cell slurry (~$10^6$ cells) was then spotted onto these plates. After 2 days of growth at 30 °C, the relative amounts of **a**-factor produced by *MATa* cells were determined from the size of the growth inhibition zone (halo) surrounding the *MATa* cells.

**Measurement of ion leakage rate.** The rate of ion leakage was measured in various *Arabidopsis* genotypes during DIS[27,37]. The 4th–6th rosette leaves were detached from 35-day-old *Arabidopsis* plants, and incubated in darkness for 0 and 5 days. Approximately three to four leaves for each condition were randomly collected in a 50-mL Falcon tube with 10 mL of ddH₂O. More than four replicates were performed for each genotype. The tubes were shaken at RT for 2 h. The initial electrolyte leakage from the leaves was determined by measuring the conductivity of the solution using a TWIN compact conductivity meter (Horiba, Kyoto, Japan). Then, the samples were boiled for 10 min and shaken at RT for 2 h. The conductivity was measured again to determine the total electrolyte leakage. The ion leakage rate was calculated from the ratio of initial to total conductivity.

**Reporting summary.** Further information on research design is available in the Nature Research Reporting Summary linked to this article.

## Data availability

All relevant data supporting the findings of this study are available within the manuscript and its supplementary files or are available from the corresponding author upon reasonable request. Raw data for underlying Figs. 1b, c, f–h, 2c–i, 3b–h, 4c, 5b, d, e, g–j, 6b, c, e–h, 7c, e–g, i–k, and Supplementary Figs. 1b, c, e–g, 2b, 3b–d, 4a, b, 5c–e, 6a–c, 7; 9c, 10b, 11b–e, 12, 14b–e; 15b, 16b, c, 17, 18b, c, 19a–c, and 20b, d–f are provided in the Source Data file.

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

## Acknowledgements

We thank Da-Peng Zhang (Tsinghua University, China) for providing CHLH antibody, Meizhong Luo (Huazhong Agricultural University, China) for providing CHLI antibody and expression constructs for MgCh subunits, Ayumi Tanaka (Hokkaido University, Japan) for providing SGR1 antibody, Benke Kuai (Fudan University, China) for providing *nye1* seeds, and Barry Pogson (Australian National University, Australia) for providing yeast strains for the yeast halo assay. This research was supported by the Alexander von Humboldt Foundation to P.W., the Elitenetzwerk Bayern to S.G. and J.R. W.K., and the Deutsche Forschungsgemeinschaft to P.W. (WA 4599/2-1), to S.G. (FOR2092, GE 1110/9-1), to B.G. (FOR2092, GR 936/18-1) B.G. and A.S.R. (SFB TRR175, subproject C04).

## Author contributions

P.W. and B.G. designed the research. P.W. performed the majority of the experiments. P.W. and A.S.R. performed in vitro MgCh assay. P.W., J.R.W.K., and S.G. performed electron microscopy analyses. P.W., A.S.R, J.R.W.K., S.G., and B.G. analyzed the data. P.W. and B.G. wrote the manuscript with contributions from all authors.

## Competing interests

The authors declare no competing interests.
