## [Peer Review File · Nature Communications]

Reviewers' comments:

Reviewer #1 (Remarks to the Author):

The present work is exploring the role of two BCM protein isoforms in the synthesis and degradation of chlorophyll molecules in higher plants. It is definitively an interesting work, providing a new insight that is of broad interest to plant biologists. Authors conclusions are based on a large amount of data, most of them logical and carefully done. Presented results show convincingly that the BCM1 protein is needed for the sufficient synthesis of MgP and the activity of this protein is somehow connected to Mg chelatase enzyme. In contrast, the BCM2 protein seems to act during senescence to slower (smooth) the process of chlorophyll breakdown. On the other hand, this ms is a bit too data-rich and not so easy to read also because of too many abbreviations used. I would recommend to remove data that are not completely necessary for the story (e.g. Suppl Fig 7, Fig. 2 could be reduced etc). I have also several concerns about the interpretation of data presented.

1. In author's model the BCM1 interacts with Gun4 and ChIM as a scaffold protein and the interaction with Gun4 directly stimulates the MgCh activity. This assumption is supported by an in vitro MgCh assay containing membranes isolated from a plant overexpressing BCM1 instead of a recombinant BCM1. A subtle stimulation is than described as an evidence that the BCM1 has an positive effect on Mg chelation. However, I think such assay is not a way how to provide a proof. There are thousands of compounds in isolated membranes and such a complex mixture can be indeed slightly different in very poor plants. It might even contain compounds oxidizing porphyrins etc. I understand the problem with detergent but than, authors could try amphipols or, possibly, isolated membranes from *E. coli* expressing BCM1 - at least, there should be much higher concentration of BCM1 for the assay (if not aggregated).

2. The model is further supported by lowered levels of some metabolites in tetrapyrrole pathway, particularly MgP and MgPME. I would not agree however that the 'ALA synthesis.. was dramatically decreased' if this decrease is just about 20%. Moreover, this value means a capacity to synthesize ALA, not a real amount of ALA produced. Could it be excluded that the pale phenotype of *bam1* mutant is caused by very low PPIX, the substrate of MgCh. By other words, could be the problem in *bcm1* upstream of MgCh? In fact, it is a bit surprising that levels of PPIX are not provided. If authors are correct, ALA feeding of *bcm1* should elevate dramatically the PPIX level and such a result can be easily incorporated into Suppl Fig. 3b.

3. Regarding the role of BCMs in chlorophyll breakdown, provided results are clear. I'm only not sure about a firm conclusion that 'BCM1 inhibit Chl breakdown independently of their function in Chl biosynthesis'. According to Fig. 1b senescent leaves still contain a detectable level of ChIM. Although the Gun4 is gone (Suppl Fig 16c), it is not clear what happens with other BCMs partners GluTR and ChIM (ChIH) in BCM1-OX line during senescence? Are these protein detectable, and if so, are their levels different from the control?

A minor comment:

I do not fully understand Co-IP figure (Fig 7c). Do + + symbols indicate that the BCM1 is co-purified with SGR1-FLAG only if the first protein is overexpressed?

Reviewer #2 (Remarks to the Author):

Wang and colleagues provide here a compelling set of data that identified two new potential regulators of the chlorophyll metabolism, BCM1 and BCM2. The experiments are well conducted

and the manuscript concise, logic and well written. This paper is an important piece of work to add to the fascinating field of chl metabolism and one of the first clear example of link between chl synthesis and degradation.

The authors show two different and partially overlapping function of two proteins of unknown function BCM1 and BCM2: BCM1 is expressed mostly in greening leaves and promotes Mg chelation via GUN4 interaction (and also prevents chl degradation), meanwhile BCM2, senescence specific, is destabilizing the Mg dechelating enzyme SGR1.

Despite the extensive and very elegant genetic data presented, the underlying mechanism of action of BCM1 and 2 remains elusive. I would like here to point out some underlying questions that remained unanswered:

- It is somehow intriguing that both BCM are apparently acting (indirectly?) at the magnesium chelation and dechelation steps in both synthesis and degradation pathway. Would that suggest a function related more directly to magnesium homeostasis? Did the authors investigate in that respect?
- Concerning the model proposed (fig8) and discussed (380 and following), while the function of BCM1 appears quite clear, I struggle to understand the actual physiological role of BCM2 in chl degradation: why an inhibitor of the pathway would be co-expressed with the enzymes (SGR, PPH, PAO...) of the pathway it is actually inhibiting? While BCM1 expression goes down during DIS, why maintaining/expressing BCM2? Could the BCM-dependent degradation of SGR (Fig. 7) only be a secondary effect of the change in Mg homeostasis?

Some other (minor) points:

- Fig.1 Why the negative control be nuclear expressed? Specify the construct used in the legend.
- Do BCM1 and BCM2 actually interacts with each other? Are other CAAX-type endopeptidase known to dimerise?
- Fig. 5c. As far as I understand this figure, the changes in MgCh activity are marginal, even if significant. gun4 activity appears overwhelming.
- Fig. 6c. It is surprising that some pheide could be detected in WT after 7dDD. This is not consistent with previous reports.
- L291. "BCM1 does not interfere with the normal course of DIS". So, why chl gets degraded quicker in bcm mutant lines? (Sfig. 12)
- L301. "conserved inhibitors of chl breakdown". We can observed an inhibition of the pathway, but they may not be really "inhibitor" in an enzymatic point of view. (Negative regulators?)
- L346. The actual stand of knowledge of SGR protein function is still under debate. It might have been nice to look into other SGR1(NYE1 paralogs (SGR2, SGRL and their potential interaction with BCM's.
- Fig. 7. Even if BCM2 could be shown as complementing bcm1 lines, and being supposedly real paralogs, it would have been more logic to look into BCM2 interaction with SGR1 (as both being coexpressed during development) in addition to BCM1 data shown here.
- L366. "evolutionaty conserved". Well, it might also be that BCM1 and 2 actually diverged in their initial function?
- L 370. Did the authors looked into proteasome or ubiquitination involvement on the various lines presented here?
- L 386. " at the onset of leaf senescence, expression of BCM2 is up-regulated to restrict the accumulation of SGR1". Why is such a control over chl degradation necessary at that development stage?

Reviewer #3 (Remarks to the Author):

In this manuscript, two new regulators of Chl biosynthesis (BCM1 and BCM2) have been identified and the authors have also conducted in-depth research on molecular mechanisms. Interestingly,

BCMs regulate the trade-off between chlorophyll synthesis and catabolism. BCMs interacts with GUN4 to regulate chlorophyll synthesis, while BCMs inhibit Chl breakdown by destabilizing SGR1. Regulation of the expression of these two genes by bioengineering has the potential to facilitate the cultivation of high-yield crops. One of the major questions is whether these two genes could help solve the problem of non-functional stay-green?

Major comments

1. The author should clarify the definition or parameters of leaf senescence in the manuscript. The most remarkable event of leaf senescence is the breakdown or degradation of chlorophyll, but NOT ALL chlorophyll breakdown could be defined as leaf senescence phenotype. A number of mutants display leaf development or chlorophyll biosynthesis defects at early stage but without any leaf senescence phenotype.
2. For Figure 1e, a clear and sharp image is required.
3. Line 988 and Figure 1e, YFP itself cannot accumulate in nucleus, and it seems that this is a NLS-YFP signal. The 35S-YFP signal should spread over the cell.
4. Figure 6a, the authors should quantify the expression of senescence associated marker genes, such as SAG12, just as in Figure 7h.
5. Co-localization assay with different fluorescence marker is needed to demonstrate that GUN4 can physically interact with BCM1, as well as that of SGR1.

Minor comments

1. Figure 3a, the double mutant 'bcm1 bcm 2' should be labeled in detail for which combination was applied because each BCM gene has two mutants, bcm1-1 and bcm 1-3 for BCM1, and bcm 2-1 and bcm 2-2 for BCM2.
2. Figure 6a, dark treatment for 7 days is a long period for Arabidopsis leaves because the Col-0 leaves became totally yellow after 7-day-dark treatment.
3. The letters of the significance 'a, b, c, d' should be uniformly marked. In Figure 2/3/4, from maximum to minimum value the authors marked 'a-d'; But in Figure 5, the authors marked 'd-a'.
4. Significance letters should be added in Figure 6c.
5. Given that authors use 'Chl' for abbreviation of 'Chlorophyll' in the text, the description should be uniform (Line 587, Line 603, Line 616, Line 627, Line 705).

Response to Reviewers:

Dear editor and reviewers,

We appreciate your positive and constructive comments and suggestions very much on our manuscript entitled “Post-translational coordination of chlorophyll biosynthesis and breakdown by BCMs maintains chlorophyll homeostasis during leaf development” (ID: #NCOMMS-19-25702). We have studied your comments and suggestions carefully and revised our manuscript according to these comments. Please find our point-by-point response below.

Reviewer #1 (Remarks to the Author):

The present work is exploring the role of two BCM protein isoforms in the synthesis and degradation of chlorophyll molecules in higher plants. It is definitively an interesting work, providing a new insight that is of broad interest to plant biologists. Authors conclusions are based on a large amount of data, most of them logical and carefully done. Presented results show convincingly that the BCM1 protein is needed for the sufficient synthesis of MgP and the activity of this protein is somehow connected to Mg chelatase enzyme. In contrast, the BCM2 protein seems to act during senescence to slower (smooth) the process of chlorophyll breakdown. On the other hand, this ms is a bit too data-rich and not so easy to read also because of too many abbreviations used. I would recommend to remove data that are not completely necessary for the story (e.g. Suppl Fig 7, Fig. 2 could be reduced etc). I have also several concerns about the interpretation of data presented.

Response: To improve the manuscript readability, we have revised the results according to your review and other reviewers’ suggestions. We performed new experiments, tried the compromise between “too data rich” and requests for additional experiments, confirmed results, modified the figure panels and organized newly the file with supplemental figures. For clarity, we attached a manuscript file with yellow labelling of all changes in the manuscript. Moreover, we refer also to the other answers on your review below.

1. In author’s model the BCM1 interacts with Gun4 and ChlM as a scaffold protein and the interaction with Gun4 directly stimulates the MgCh activity. This assumption is supported by an in vitro MgCh assay containing membranes isolated from a plant overexpressing BCM1 instead of a recombinant BCM1. A subtle stimulation is than described as an evidence that the BCM1 has an positive effect on Mg chelation. However, I think such assay is not a way how to provide a proof. There are thousands of compounds in isolated membranes and such a complex mixture can be indeed slightly different in very poor plants. It might even contain compounds oxidizing porphyrins etc. I understand the problem with detergent but than, authors could try amphipols or, possibly, isolated membranes from E. coli expressing BCM1 - at least, there should be much higher concentration of BCM1 for the assay (if not aggregated).

Response: We have tried to express and purify recombinant His-BCM1 in *E. coli*, however, BCM1 will form detergent-resistant inclusion bodies. One explanation could be that efficient expression of heterologous membrane-localized BCM1 in *E. coli* will result in its misfolding and aggregation¹. Instead, we succeeded to express recombinant His-BCM1 in *Saccharomyces cerevisiae* cells and subsequently purify His-BCM1 in the presence of β -DM. So, we first performed MgCh assay by using isolated yeast membranes in the presence or

absence of BCM1. We found that the use of BCM1-containing yeast membranes increases MgCh activity by 12% compared to a MgCh assay supplemented with yeast membranes without BCM1 (Supplementary Fig. 6a). Secondly, it was expected that the addition of higher concentration of purified BCM1 could have a stronger effect on MgCh activity. We found that the use of 1 μ M His-BCM1 increases MgCh activity by \sim 3-fold compared to the assays without BCM1 or supplemented with 1 μ M GST (Supplementary Fig. 6c). All in all, our results suggest that BCM1 is a positive and supportive regulator of MgCh activity. In addition, it might be a good, but challenging idea to try amphipol or nanodisc reconstitution of the purified BCM1 and verify then the MgCh assays in the near future. But due to a successful purification of sufficient amount of His-BCM, these approaches seem not to be too urgent.

2. The model is further supported by lowered levels of some metabolites in tetrapyrrole pathway, particularly MgP and MgPME. I would not agree however that the 'ALA synthesis. was dramatically decreased' if this decrease is just about 20%. Moreover, this value means a capacity to synthesize ALA, not a real amount of ALA produced. Could it be excluded that the pale phenotype of bam1 mutant is caused by very low PPIX, the substrate of MgCh. By other words, could be the problem in bcm1 upstream of MgCh? In fact, it is a bit surprising that levels of PPIX are not provided. If authors are correct, ALA feeding of bcm1 should elevate dramatically the PPIX level and such a result can be easily incorporated into Suppl Fig. 3b.

Response: We have revised the description of the results of ALA synthesis rate in the manuscript. We have tried to measure the steady-state levels of ALA, however, the amount of ALA was generally too low to be detectable. So, we supplemented levulinic acid, the inhibitor of ALA dehydratase, to the incubation buffer to allow the accumulation of ALA for a fixed time period. To exclude the possibility that the pale-green phenotype of *bcm1* mutants is caused by very low Proto, we included in the revised manuscript the steady-state levels of Proto in WT, *bcm1* and BCM1-OX seedlings and found that Proto in *bcm1* was decreased by 10% compared to that in WT seedlings (Fig. 2d). We prefer to add this result in the main figure rather than in a supplementary figure. The results suggest that the drastically reduced content of Mg-porphyrins observed in *bcm1* could not be explained by a reduced content of Proto. In this context, it is interesting to mention that the *bcm1* mutant as well as the other MgCh subunits deficient mutants, such as *gun4* and *gun5*, did not accumulate higher amounts of Proto compared to WT, although Proto is the substrate of MgCh (Fig. 2d)^{2,3}. These observations suggest the existence of an essential feedback mechanism triggered by reduced MgCh activity, that prevents the accumulation of phototoxic Proto^{4,6}.

3. Regarding the role of BCMs in chlorophyll breakdown, provided results are clear. I'm only not sure about a firm conclusion that 'BCM's inhibit Chl breakdown independently of their function in Chl biosynthesis'. According to Fig. 1b senescent leaves still contain a detectable level of ChlM. Although the Gun4 is gone (Suppl Fig 16c), it is not clear what happens with other BCM's partners GluTR and ChlM (ChlH) in BCM1-OX line during senescence? Are these proteins detectable, and if so, are their levels different from the control?

Response: We included in the revised version manuscript the steady-state levels of BCMs' interaction partners, including GluTR, GUN4, CHLH and CHLM during dark incubation. Our results suggest that all these TBS proteins are drastically decreased during dark incubation and are undetectable at 3 DDI (Supplementary Fig. 17).

A minor comment:

I do not fully understand Co-IP figure (Fig 7c). Do ++ symbols indicate that the BCM1 is co-purified with SGR1-FLAG only if the first protein is overexpressed?

Response: The Co-IP analyses were conducted as described previously^{7,8}. The “++” symbols mean that both BCM1 and SGR1-FLAG were overexpressed in the same *Nicotiana benthamiana* leaves for two days. The chloroplasts were then isolated from tobacco leaves and used for the immunoprecipitation assay by using anti-FLAG affinity beads. The co-purification of BCM1 with SGR1-FLAG suggests that both proteins interact with each other.

Reviewer #2 (Remarks to the Author):

Wang and colleagues provide here a compelling set of data that identified two new potential regulators of the chlorophyll metabolism, BCM1 and BCM2. The experiments are well conducted and the manuscript concise, logic and well written. This paper is an important piece of work to add to the fascinating field of chl metabolism and one of the first clear example of link between chl synthesis and degradation. The authors show two different and partially overlapping function of two proteins of unknown function BCM1 and BCM2: BCM1 is expressed mostly in greening leaves and promotes Mg chelation via GUN4 interaction (and also prevents chl degradation), meanwhile BCM2, senescence specific, is destabilizing the Mg dechelating enzyme SGR1. Despite the extensive and very elegant genetic data presented, the underlying mechanism of action of BCM1 and 2 remains elusive. I would like here to point out some underlying questions that remained unanswered:

1. It is somehow intriguing that both BCM are apparently acting (indirectly?) at the magnesium chelation and dechelation steps in both synthesis and degradation pathway. Would that suggest a function related more directly to magnesium homeostasis? Did the authors investigate in that respect?

Response: That is potentially a good idea. But we did not go in this direction and did not perform further experiments for the moment. One reason is that the mutants with defect in magnesium homeostasis show a yellow reticulated vein phenotype⁹⁻¹¹, which was not observed in *bcm1* mutants (Figs. 2, 3 and 5). Another reason is: If BCMs are indeed directly involved in the regulation of magnesium homeostasis, we would expect that BCMs are mainly localized in the envelope membrane of chloroplast⁹. However, we found BCM1 is mainly localized in the thylakoid membranes (Fig. 1f). Nevertheless, it was recently reported that SGR-mediated chlorophyll degradation is involved in the remobilization of magnesium¹². Thus, it cannot be excluded that the opposite effect of BCMs on MgCh and SGR could indirectly or directly affect the remobilization and/or distribution of magnesium within the chloroplast or between source tissue and sink tissue.

2. Concerning the model proposed (fig8) and discussed l380 and following), while the function of BCM1 appears quite clear, I struggle to understand the actual physiological role of BCM2 in chl degradation: why an inhibitor of the pathway would be co-expressed with the enzymes (SGR, PPH, PAO...) of the pathway it is actually inhibiting? While BCM1 expression goes down during DIS, why maintaining/expressing BCM2? Could the BCM-dependent degradation of SGR (Fig. 7) only be a secondary effect of the change in Mg homeostasis?

Response: Our explanation is that up-regulation of *BCM2* at the onset of leaf senescence can compensate for the dysfunction of *BCM1*, which is down-regulated during senescence. The significance of cooperative function of two BCM isoforms is highlighted by the preferentially enhanced pale-green phenotype of old leaves rather than young leaves of *bcm1 bcm2* double silencing plants compared to *bcm1* single mutant and WT (Fig. 5a-e). Consistently, we also observed a faster chlorophyll degradation during dark incubation in the *bcm1-3 bcm2-2* double mutant compared to WT (Fig. 6a-d). It is worth mentioning that the stability of *BCM2* protein was decreased during dark incubation (Fig. 6g), although *BCM2* is up-regulated during senescence (Supplementary Fig. 9c). This correlates with accumulation of SGR1 and thus with a boost in the rate of chlorophyll breakdown at the late stage of senescence. In this context, we propose a working model for the two BCMs: *BCM1* is the predominant isoform during vegetative growth to stimulate MgCh activity and optimize chlorophyll biosynthesis. During leaf maturation and the transition to leaf senescence, the two BCM isoforms cooperatively to restrict the accumulation of SGR to achieve a smooth transition to chlorophyll breakdown at the beginning of leaf senescence. In addition, we agree that it remains to be addressed in future whether alteration of Mg homeostasis affect Chl breakdown⁹.

Some other (minor) points:

1. Fig.1 Why the negative control be nuclear expressed? Specify the construct used in the legend.

Response: The GFP/YFP proteins are mainly localized in cytosol and can diffuse into nucleus¹³⁻¹⁷. We have added new results in Fig. 1e and revised the figure legend.

2. Do *BCM1* and *BCM2* actually interacts with each other? Are other CAAX-type endopeptidases known to dimerise?

Response: To answer these questions, we conducted yeast two-hybrid (Y2H) experiments to test the interaction between two BCMs. We found that both BCMs can interact with each other (Fig. 1 in the Response letter). There is no published data showing the dimerization of CAAX-type endopeptidase^{8,18-20}. In addition, our results do not support the CAAX-type peptidase function of two BCMs, as expression of BCMs was not able to rescue a CAAX-peptidase-deficient yeast mutant strain (*rce1Δ ste24Δ*) (Supplementary Fig. 10a). The two BCMs have no conserved motif for a peptidase (Supplementary Fig. 9a). Since the physiological significance of dimerization of BCMs in vivo remains to be addressed, we suggest leaving out the results on BCM's dimerization in the manuscript (Fig. 4a).

Fig. 1. Yeast-two-hybrid analyses of protein interaction between two BCMs.

The transformed yeast strains were analyzed on selective medium lacking Leu and Trp (SD/-L-T) or His, Leu, Trp and Ura (SD/-H-L-T-U) in the presence of 25 mM 3-amino-1,2,4-triazole (3-AT). The NubG-GSAT and Cub-GSAT were used as the negative control for NubG-BCM1 and Cub-BCM1, respectively.

3. Fig. 5c. As far as I understand this figure, the changes in MgCh activity are marginal, even if significant. *gun4* activity appears overwhelming.

Response: Because no known catalytic domain is found in BCM1 structure, it is proposed that BCM1 is not an enzyme, but most likely a membrane anchor, and assists in the assembly and organization of Chl synthesis in the chloroplast. We performed new MgCh assays with and without BCM1 (Fig. 4c and Supplementary Fig. 6). We refer to our response to major question #1 of reviewer #1 on Page 2.

Our in vivo and in vitro results (Figs. 2-4) suggest that BCM1 acts as scaffold for the transient membrane contact of the active GUN4-MgCh complex. Thus, BCM1 modulates the activity of MgCh for the Mg chelation of Proto. But we do not expect a strong stimulation of MgCh activity in vitro by BCM1 as GUN4 acts positively on Mg chelation of Proto, while BCM1 is a supportive auxiliary or assembly factor. In fact, GUN4 is suggested to be capable to bind Proto and MgP, the substrate and product of MgCh, respectively²¹. Thus, GUN4 greatly stimulates MgCh activity most likely by interaction with substrate-binding and catalytic CHLH subunit and could provide the substrates and/or release the product of MgCh^{22,23}. In consequence, we and other laboratories observed the seedling-lethal phenotype of *gun4* knock-out mutant under photoperiodic growth conditions^{21,24}. And we observed that *gun4-1* (a *gun4* knock-down mutant) showed a much stronger pale-green leaf phenotype than the *bcm1* mutants do (Fig. 3a), implying that GUN4 acts decisively on the Mg chelation step, which is different from the role of BCM1.

Furthermore, it was not excluded that the addition of higher amounts of purified BCM1 could even have a stronger stimulated effect on MgCh activity. To test this hypothesis, we expressed His-BCM1 in *Saccharomyces cerevisiae* cells and successfully purified His-BCM1 in the presence of β -DM. We found that the addition of 1 μ M purified His-BCM1 increase MgCh activity by \sim 3-fold compared to the assays without BCM1 or supplemented with 1 μ M GST (Supplementary Fig. 6c). All in all, our results suggest that BCM1 is a positive and supportive regulator of MgCh activity in the membrane.

4. Fig. 6c. It is surprising that some pheide could be detected in WT after 7dDD. This is not consistent with previous reports.

Response: Our reproducible results indicate that Pheide *a* could be detectable at 7 DDI, but not at 5 DDI (Fig. 2c in the Response letter). The huge amount of Pheide *a* observed in *pao* mutant is a good prove for our HPLC-based measurement of Pheide *a*²⁵ (Fig. 2c in the Response letter). Two aspects may explain the difference in Pheide *a* measurement between our results and previous reports. Firstly, to unify the growth condition for both Chl biosynthesis and breakdown analyses, we grew *Arabidopsis* plants under short-day (10 h light/14 h dark) condition rather than long-day (16 h light/8 h dark). As a result, the rate of Chl degradation and accumulation of Chl catabolites in the detached leaves will be different. Secondly, our pigment/Chl catabolite extraction method and HPLC protocol could also lead to certain differences. More details are described in the section of Methods.

Fig. 2. BCM-OX plants exhibit stay-green leaf phenotype during dark incubation.

a Representative images of detached leaves from 35-day-old *Arabidopsis* plants grown under short-day normal light ($120 \mu\text{mol photons m}^{-2} \text{s}^{-1}$) conditions prior to (0 DDI) and after 5 or 7 days of dark incubation (5 or 7 DDI). Scale bars, 0.5 cm. **b** and **c** Levels of Chl(**b**) and Chl catabolites (**c**) in the detached leaves shown in (**a**) at 0, 5 and 7 DDI.

5. L291. “*BCM1* does not interfere with the normal course of *DIS*”. So, why chl gets degraded quicker in *bcm* mutant lines? (Sfig. 12)

Response: We have revised this sentence in the manuscript for the sake of clarity. We intend to suggest that the overexpression of BCM specifically inhibits the rate of Chl breakdown during dark incubation.

6. L301. “conserved inhibitors of chl breakdown”. We can observe an inhibition of the pathway, but they may not be really “inhibitor” in an enzymatic point of view. (Negative regulators?)

Response: We agree with your comment. We have replaced “inhibitor” by “negative regulator” in the manuscript.

7. L346. The actual stand of knowledge of SGR protein function is still under debate. It might have been nice to look into other SGR1(NYE1 paralogs (SGR2, SGRL and their potential interaction with BCM's.

Response: To answer this question, we carried out the Y2H assay to test the interaction of BCMs with three SGR isoforms in *Arabidopsis*. The results suggest that both BCMs can interact with both SGR1 and SGR2, but not with SGRL (Fig. 3). Considering the fact that the *SGRL* is predominantly expressed during the vegetative growth and gradually down-regulated as senescence induces the expression of *SGR1* and *SGR2*^{26,27}, we suggest that BCMs plays an essential role in the control of Chl breakdown by their interaction with SGR1/2. Since SGR1 was suggested to be the predominant isoform during senescence^{28,29}, we favor to apply SGR1 as the main representative target of BCMs in this manuscript. Beside the difference in gene expression during leaf development and senescence among the three SGR paralogs, SGR1/2 and SGRL revealed a different substrate specificity in vitro³⁰. SGRL showed a higher activity to release Mg from Chlide *a* than from Chl *a*. The molecular mechanism behind the selective protein interaction of BCMs with the SGR paralogs needs more detailed study in the near future. Due to the complexity of the first report on the role of BCMs in chlorophyll metabolism and for improved legibility of the manuscript, we hesitate to include additionally these results to the manuscript.

Fig. 3. Y2H analyses of protein interaction of BCMs with SGR homologs in *Arabidopsis*.

The transformed yeast strains were analyzed on selective medium lacking Leu and Trp (SD/-L-T) or His, Leu, Trp and Ura (SD/-H-L-T-U) in the presence of 25 mM 3-AT. The NubG-GSAT and Cub-GSAT were used as the negative control for NubG-BCM and Cub-SGR, respectively.

8. Fig. 7. Even if BCM2 could be shown as complementing *bcm1* lines, and being supposedly real paralogs, it would have been more logic to look into BCM2 interaction with SGR1 (as both being coexpressed during development) in addition to BCM1 data shown here.

Response: The interaction between BCM2 and SGR1 is shown in the Supplementary Fig. 20.

9. L366. “evolutionary conserved”. Well, it might also be that BCM1 and 2 actually diverged in their initial function?

Response: Derived from the phylogenetic tree, we found BCM paralogs are exclusively found in land plants (Supplementary Fig. 8). Different plant species have various amounts of BCM paralogs. For example, one BCM paralog is found in *Oryza sativa* and *Solanum lycopersicum*, while two paralogs are found in *Arabidopsis thaliana* and *Glycine max*. It has been shown that deficiency of BCM paralogs in *Arabidopsis thaliana*, *Oryza*

sativa and *Glycine max* led to pale-green leaf phenotype and seed coat⁸. It is suggested that BCM paralogs play an evolutionarily conserved role in Chl metabolism among different land plants. Our results emphasize both BCMs in *Arabidopsis* act as conserved regulatory factors in Chl metabolism. Owing to their distinct expression activity in young green leaves and senescent leaves (Supplementary Fig. 9b, c), two BCMs are suggested to modulate the balance between Chl synthesis and Chl breakdown.

10. L 370. Did the authors look into proteasome or ubiquitination involvement on the various lines presented here?

Response: We have not conducted a proteasome approach. The *bcm1 bcm2* double silencing plants showed enhanced pale-green phenotype of aged leaves (Fig. 5a, c) and would be in future good plant material for proteomic analyses. Protein degradation was an issue for our studies on BCM, but we did not see specific effects of BCM stability by using some mutants for plastid proteases (such as *clp* protease subunits and *ftsh* subunits). It remains to be determined whether protein ubiquitination exists in chloroplast^{18,31,32}.

11. L 386. “at the onset of leaf senescence, expression of BCM2 is up-regulated to restrict the accumulation of SGR1”. Why is such a control over chl degradation necessary at that development stage?

Response: We refer to our response to major question #2 of reviewer #2 on Page 4.

Reviewer #3 (Remarks to the Author):

In this manuscript, two new regulators of Chl biosynthesis (BCM1 and BCM2) have been identified and the authors have also conducted in-depth research on molecular mechanisms. Interestingly, BCMs regulate the trade-off between chlorophyll synthesis and catabolism. BCMs interacts with GUN4 to regulate chlorophyll synthesis, while BCMs inhibit Chl breakdown by destabilizing SGR1. Regulation of the expression of these two genes by bioengineering has the potential to facilitate the cultivation of high-yield crops. One of the major questions is whether these two genes could help solve the problem of non-functional stay-green?

Response: That is an intriguing point. We are aware that BCM1/2-OX plants just exhibit the classic non-functional stay-green phenotype during dark incubation. Although PS-LHC complexes were much more stable in BCM1-OX plants than in WT plants during dark incubation, cytochrome b₆f and ATP synthase complexes were degraded in BCM1-OX seedlings during DIS, similar to WT and *bcm1-3* seedlings under the same conditions (Supplementary Fig. 15). It is rather unlikely to envision a sustained photosynthesis in BCM1/2-OX plants after long-term dark treatment. Nevertheless, the photosynthetic efficiency and plant fitness remain to be determined in BCM1/2-OX plants in response to various biotic and abiotic stresses, such as pathogen infection, changing light intensities and water deprivation in future experiments³¹.

Major comments:

1. The author should clarify the definition or parameters of leaf senescence in the manuscript. The most remarkable event of leaf senescence is the breakdown or degradation of chlorophyll, but NOT ALL chlorophyll breakdown could be defined as leaf senescence phenotype. A number of mutants display leaf development or chlorophyll biosynthesis defects at early stage but without any leaf senescence phenotype.

Response: We absolutely agree with your comments. In our manuscript, we have included essential parameters of leaf senescence in the manuscript, such as degradation of Chl and proteins, up-regulation of senescence associated gene (*SAG12*) and Chl catabolic genes (*NYC1*, *SGR1*, *PPH* and *PAO*), and increased ion leakage of detached leaf during dark incubation (Fig. 6 and Supplementary Figs. 14, 15, 19). Our results suggest that deficiency or overproduction of BCMs specifically and exclusively change Chl degradation rate during dark incubation.

In particular, we suggest that the pale-green leaf phenotype of *bcm1* mutant can be observed in both young and old leaves due to the defect in Chl biosynthesis and is not related to modified senescence. In contrast, the preferentially enhanced pale-green phenotype of old leaves rather than young leaves of *bcm1 bcm2* double silencing plants (Fig. 5a-e) is a good hint for accelerated Chl breakdown occurred in aged leaves, but not a hint for early leaf senescent phenotype, because both old and young leaves of *bcm1 bcm2* double silencing plants showed similar expression of *SAG12* and *SGR1* (Supplementary Fig. 12). All in all, we propose a working model in which both BCMs coordinate Chl biosynthesis and breakdown during leaf development and senescence. The potential function of BCMs in the control of senescence under normal growth conditions or stress conditions need to be investigated in the future.

2. For Figure 1e, a clear and sharp image is required.

Response: We have added high resolution images in Fig. 1e.

3. Line 988 and Figure 1e, YFP itself cannot accumulate in nucleus, and it seems that this is a NLS-YFP signal. The 35S-YFP signal should spread over the cell.

Response: We have repeated the experiments in Fig. 1e. The outcome is consistent with previous reports¹³⁻¹⁷, GFP/YFP proteins mainly localized in cytosol and can diffuse into nucleus. Importantly, both BCM1-YFP and cTP_{BCM1}-YFP fusion proteins are found to be localized in chloroplasts, suggesting the chloroplast localization of BCM1. So, we have added new results in Fig. 1e and revised the figure legend.

4. Figure 6a, the authors should quantify the expression of senescence associated marker genes, such as *SAG12*, just as in Figure 7h.

Response: We have added the expression of *SAG12* and *SGR1* in Fig. 6e, f. The results suggest both genes are WT-likely up-regulated in the *bcm1-3 bcm2-2* double mutant and BCM1/2-OX lines during dark incubation.

5. Co-localization assay with different fluorescence marker is needed to demonstrate that *GUN4* can physically interact with BCM1, as well as that of *SGR1*.

Response: To determine the co-localization of BCM1 with *GUN4* or *SGR1*, we transformed the *pUC-BCM1-YFP* together with *pUC-GUN4-CFP* or *pUC-SGR1-CFP* plasmids into *Arabidopsis* mesophyll protoplasts as described previously¹³. After 14 h of incubation in darkness, we found that BCM1-YFP fluorescence can overlay with *GUN4/SGR1-CFP* fluorescence (Fig. 4 in the response letter), confirming chloroplast targeting of these proteins and suggesting multiple interaction partners of BCM1 in chloroplast. Consistently, beside *GUN4* and *SGR1*, nine-cis-epoxycarotenoid dioxygenase 3 (NCED3) and phytoene synthase (PSY) also interact with

BCM1⁸. As BiFC results directly verify the protein-protein interaction and co-localization of BCM1 with GUN4 and SGR1 (Figs. 4b and 7b), we suggest keeping BiFC results in the manuscript.

Fig. 4. Co-localization analyses in *Arabidopsis* protoplast.

Subcellular localization of the BCM1-YFP together with GUN4-CFP or SGR1-CFP was analysed using excitation/emission filters for YFP fluorescence (Ex/Em, 514/530-555 nm) and CFP fluorescence (Ex/Em, 440/450-500 nm) with a confocal laser-scanning microscope Leica TCS SP2 (Leica Microsystems, Wetzlar, Germany). Scale bars, 5 μ m.

Minor comments:

1. Figure 3a, the double mutant '*bcm1 bcm 2*' should be labeled in detail for which combination was applied because each BCM gene has two mutants, *bcm1-1* and *bcm 1-3* for BCM1, and *bcm 2-1* and *bcm 2-2* for BCM2.

Response: These points have been revised in the manuscript.

2. Figure 6a, dark treatment for 7 days is a long period for *Arabidopsis* leaves because the *Col-0* leaves became totally yellow after 7-day-dark treatment.

Response: In our dark incubation experiments, we have carefully monitored Chl degradation rates at a series of dark incubation time points, such as 3, 5 and 7 days of dark incubation shown in Supplementary Fig. 14. For Fig. 6a, we also conducted dark treatment for 5 days (Fig. 2 in the Response letter). We observed similar differences in the amounts of Chl and its catabolites at 5 and 7 DDI. To improve clarity and readability of the manuscript, we only showed the results at 7 DDI.

3. The letters of the significance '*a, b, c, d*' should be uniformly marked. In Figure 2/3/4, from maximum to minimum value the authors marked '*a-d*'; But in Figure 5, the authors marked '*d-a*'.

Response: We have unified the sequence of significance letters in the manuscript.

4. Significance letters should be added in Figure 6c.

Response: The significance letters have been added in the manuscript.

5. Given that authors use 'Chl' for abbreviation of 'Chlorophyll' in the text, the description should be uniform (Line 587, Line 603, Line 616, Line 627, Line 705).

Response: These points have been revised in the manuscript.

References

- 1 Bernaudat, F. *et al.* Heterologous expression of membrane proteins: choosing the appropriate host. *PLoS One* **6**, e29191, doi:10.1371/journal.pone.0029191 (2011).
- 2 Mochizuki, N., Brusslan, J. A., Larkin, R., Nagatani, A. & Chory, J. Arabidopsis genomes uncoupled 5 (GUN5) mutant reveals the involvement of Mg-chelatase H subunit in plastid-to-nucleus signal transduction. *Proceedings of the National Academy of Sciences of the United States of America* **98**, 2053-2058, doi:10.1073/pnas.98.4.2053 (2001).
- 3 Schlicke, H. *et al.* Induced deactivation of genes encoding chlorophyll biosynthesis enzymes disentangles tetrapyrrole-mediated retrograde signaling. *Mol Plant* **7**, 1211-1227, doi:10.1093/mp/ssu034 (2014).
- 4 Tanaka, R. & Tanaka, A. Tetrapyrrole biosynthesis in higher plants. *Annu Rev Plant Biol* **58**, 321-346, doi:10.1146/annurev.arplant.57.032905.105448 (2007).
- 5 Czarnecki, O. & Grimm, B. Post-translational control of tetrapyrrole biosynthesis in plants, algae, and cyanobacteria. *J Exp Bot* **63**, 1675-1687, doi:10.1093/jxb/err437err437 [pii] (2012).
- 6 Brzezowski, P., Richter, A. S. & Grimm, B. Regulation and function of tetrapyrrole biosynthesis in plants and algae. *Biochim Biophys Acta* **1847**, 968-985, doi:10.1016/j.bbabi.2015.05.007S0005-2728(15)00088-2 [pii] (2015).
- 7 Wang, P. *et al.* Chloroplast SRP43 acts as a chaperone for glutamyl-tRNA reductase, the rate-limiting enzyme in tetrapyrrole biosynthesis. *Proceedings of the National Academy of Sciences of the United States of America* **115**, E3588-E3596, doi:10.1073/pnas.1719645115 (2018).
- 8 Wang, M. *et al.* Parallel selection on a dormancy gene during domestication of crops from multiple families. *Nat Genet* **50**, 1435-1441, doi:10.1038/s41588-018-0229-2 (2018).
- 9 Tang, R. J. & Luan, S. Regulation of calcium and magnesium homeostasis in plants: from transporters to signaling network. *Curr Opin Plant Biol* **39**, 97-105, doi:10.1016/j.pbi.2017.06.009 (2017).
- 10 Liang, S. *et al.* Mutations in the Arabidopsis AtMRS2-11/AtMGT10/VAR5 Gene Cause Leaf Reticulation. *Front Plant Sci* **8**, 2007, doi:10.3389/fpls.2017.02007 (2017).
- 11 Sun, Y., Yang, R., Li, L. & Huang, J. The Magnesium Transporter MGT10 Is Essential for Chloroplast Development and Photosynthesis in Arabidopsis thaliana. *Mol Plant* **10**, 1584-1587, doi:10.1016/j.molp.2017.09.017 (2017).
- 12 Peng, Y. Y. *et al.* Magnesium Deficiency Triggers SGR-Mediated Chlorophyll Degradation for Magnesium Remobilization. *Plant physiology* **181**, 262-275, doi:10.1104/pp.19.00610 (2019).
- 13 Yoo, S. D., Cho, Y. H. & Sheen, J. Arabidopsis mesophyll protoplasts: a versatile cell system for transient gene expression analysis. *Nat Protoc* **2**, 1565-1572, doi:10.1038/nprot.2007.199 (2007).
- 14 Zhai, Z., Sooksa-nguan, T. & Vatamaniuk, O. K. Establishing RNA interference as a reverse-genetic approach for gene functional analysis in protoplasts. *Plant physiology* **149**, 642-652, doi:10.1104/pp.108.130260 (2009).
- 15 Zhang, Y. *et al.* A highly efficient rice green tissue protoplast system for transient gene expression and studying light/chloroplast-related processes. *Plant methods* **7**, 30, doi:10.1186/1746-4811-7-30 (2011).
- 16 Jin, H. *et al.* HYPERSENSITIVE TO HIGH LIGHT1 interacts with LOW QUANTUM YIELD OF PHOTOSYSTEM III and functions in protection of photosystem II from photodamage in Arabidopsis. *The Plant cell* **26**, 1213-1229, doi:10.1105/tpc.113.122424 (2014).
- 17 Jin, H. *et al.* LOW PHOTOSYNTHETIC EFFICIENCY 1 is required for light-regulated photosystem II biogenesis in Arabidopsis. *Proceedings of the National Academy of Sciences of the United States of America* **115**, E6075-E6084, doi:10.1073/pnas.1807364115 (2018).
- 18 van Wijk, K. J. Protein maturation and proteolysis in plant plastids, mitochondria, and peroxisomes. *Annu Rev Plant Biol* **66**, 75-111, doi:10.1146/annurev-arplant-043014-115547 (2015).

- 19 Cadinanos, J. *et al.* AtFACE-2, a functional prenylated protein protease from *Arabidopsis thaliana* related to mammalian Ras-converting enzymes. *The Journal of biological chemistry* **278**, 42091-42097, doi:10.1074/jbc.M306700200M306700200 [pii] (2003).
- 20 Bracha, K., Lavy, M. & Yalovsky, S. The *Arabidopsis* AtSTE24 is a CAAX protease with broad substrate specificity. *The Journal of biological chemistry* **277**, 29856-29864, doi:10.1074/jbc.M202916200 (2002).
- 21 Larkin, R. M., Alonso, J. M., Ecker, J. R. & Chory, J. GUN4, a regulator of chlorophyll synthesis and intracellular signaling. *Science* **299**, 902-906, doi:10.1126/science.1079978 (2003).
- 22 Axelsson, E. *et al.* Recessiveness and dominance in barley mutants deficient in Mg-chelatase subunit D, an AAA protein involved in chlorophyll biosynthesis. *The Plant cell* **18**, 3606-3616, doi:10.1105/tpc.106.042374 (2006).
- 23 Masuda, T. Recent overview of the Mg branch of the tetrapyrrole biosynthesis leading to chlorophylls. *Photosynth Res* **96**, 121-143, doi:10.1007/s11120-008-9291-4 (2008).
- 24 Richter, A. S. *et al.* Phosphorylation of GENOMES UNCOUPLED 4 Alters Stimulation of Mg Chelatase Activity in Angiosperms. *Plant physiology* **172**, 1578-1595, doi:10.1104/pp.16.01036 (2016).
- 25 Pruzinska, A. *et al.* Chlorophyll breakdown in senescent *Arabidopsis* leaves. Characterization of chlorophyll catabolites and of chlorophyll catabolic enzymes involved in the degreening reaction. *Plant physiology* **139**, 52-63, doi:10.1104/pp.105.065870 (2005).
- 26 Sakuraba, Y. *et al.* *Arabidopsis* STAY-GREEN2 is a negative regulator of chlorophyll degradation during leaf senescence. *Mol Plant* **7**, 1288-1302, doi:10.1093/mp/ssu045 (2014).
- 27 Sakuraba, Y., Kim, D., Kim, Y. S., Hortensteiner, S. & Paek, N. C. *Arabidopsis* STAYGREEN-LIKE (SGRL) promotes abiotic stress-induced leaf yellowing during vegetative growth. *FEBS Lett* **588**, 3830-3837, doi:10.1016/j.febslet.2014.09.018 (2014).
- 28 Wu, S. *et al.* NON-YELLOWING2 (NYE2), a Close Paralog of NYE1, Plays a Positive Role in Chlorophyll Degradation in *Arabidopsis*. *Mol Plant* **9**, 624-627, doi:10.1016/j.molp.2015.12.016 (2016).
- 29 Kuai, B., Chen, J. & Hortensteiner, S. The biochemistry and molecular biology of chlorophyll breakdown. *J Exp Bot* **69**, 751-767, doi:10.1093/jxb/erx322 (2018).
- 30 Shimoda, Y., Ito, H. & Tanaka, A. *Arabidopsis* STAY-GREEN, Mendel's Green Cotyledon Gene, Encodes Magnesium-Dechelatease. *The Plant cell* **28**, 2147-2160, doi:10.1105/tpc.16.00428 (2016).
- 31 Friso, G. & van Wijk, K. J. Posttranslational Protein Modifications in Plant Metabolism. *Plant physiology* **169**, 1469-1487, doi:10.1104/pp.15.01378 (2015).
- 32 Nishimura, K., Kato, Y. & Sakamoto, W. Essentials of Proteolytic Machineries in Chloroplasts. *Mol Plant* **10**, 4-19, doi:10.1016/j.molp.2016.08.005 (2017).

REVIEWERS' COMMENTS:

Reviewer #1 (Remarks to the Author):

My comments have been fully addressed in the revised version of this manuscript. Stimulation of the Mg chelatase activity by BCM1 has been clearly demonstrated by a strong positive effect of the recombinant BCM1 on the formation of MgP (Suppl Fig 6c). I think that this new result is much more significant than the subtle stimulation of the Mg chelatase by BCM1-enriched membranes (Fig. 4c, Suppl Fig. 6a). I would therefore suggest to replace the Fig. 4c with the Suppl Fig 6c. It is nonetheless pity that authors do not provide a control assay containing recombinant BCM1 but lacking GUN4 as they do for in vitro assays containing BCM1-enriched membranes. Such experiment would show more convincingly than current Fig. 4c and Suppl Fig. 6a whether the stimulation of Mg chelatase in vitro is provided exclusively by an interplay between BCM1 and GUN4.

Reviewer #2 (Remarks to the Author):

I would like to thank the authors for taking into account most of our comments. I believe this work is a significant advance in our understanding of the chl metabolism. However, I would still like to point out that the proposed model is somehow counter-intuitive (point 3 been made previously). I still believe that based on the data presented, BCM2 function remains somehow unclear. This is fine, given the novelty of the concepts (and the large amount of data) provided in the paper, but I would have like to have a little more in the Discussion explaining better the rationale for BCM2 function in chl degradation. There is no obvious need for a down-regulation of the one-way chl degradation pathway during leaf senescence.

Reviewer #3 (Remarks to the Author):

In the revised manuscript, most of the questions had been answered. However, there are still some minor issues that need to be addressed further.

1. The quality of Fig. 6a needs to be improved. The resolution of this image is not high enough.
2. I suggest the authors to change the Arabic numerals in these two paragraphs (lines 336-339; lines 398-403) to Roman numerals to make the paper more readable.
3. Transgenic plants should be shown in italics.

Response to Reviewers:

Dear editor and reviewers,

We appreciate your positive and constructive comments and suggestions on our manuscript entitled “Post-translational coordination of chlorophyll biosynthesis and breakdown by BCMs maintains chlorophyll homeostasis during leaf development” (ID: #NCOMMS-19-25702A). We have carefully studied your comments and suggestions and revised our manuscript according to these comments. Please find our point-by-point response below.

Reviewer #1 (Remarks to the Author):

My comments have been fully addressed in the revised version of this manuscript. Stimulation of the Mg chelatase activity by BCM1 has been clearly demonstrated by a strong positive effect of the recombinant BCM1 on the formation of MgP (Suppl Fig 6c). I think that this new result is much more significant than the subtle stimulation of the Mg chelatase by BCM1-enriched membranes (Fig. 4c, Suppl Fig. 6a). I would therefore suggest to replace the Fig. 4c with the Suppl Fig 6c.

Response: As BCMs are integral membrane protein, we suggest to analyze the effect of BCM1 on MgCh activity when BCM1 is present in the membrane. Considering the fact that the use of BCM1-containing thylakoid membranes and yeast membranes have similar stimulatory effect on MgCh activity in vitro (Fig. 4c and Supplementary Fig. 6a), we suggest to present the results obtained with thylakoid membranes as the main figure. Fig. 4c highlights the MgCh assay at the thylakoid membrane, the compartment of BCMs, while Supplementary. Fig. 6c shows the MgCh activity exclusively with recombinant proteins (without membranes). However, as we emphasize, BCM acts on membranes.

In this context, we refer to our previous response to the reviewer’s questions on the MgCh enzymatic assay. Our in vivo and in vitro results (Figs. 2-4) suggest that BCM1 acts as scaffold for the transient membrane contact of the active GUN4-MgCh complex. Thus, BCM1 modulates the activity of MgCh for the Mg chelation of Proto. But we do not expect a strong stimulation of MgCh activity in vitro by BCM1 as GUN4 acts positively on Mg chelation of Proto, while BCM1 is a supportive auxiliary or assembly factor for the attachment of MgCh complex to the thylakoid membrane. In fact, GUN4 is suggested to be capable to bind Proto and MgP, the substrate and product of MgCh, respectively¹. Thus, GUN4 greatly stimulates MgCh activity most likely by interaction with substrate-binding and catalytic CHLH subunit and could provide the substrates and/or release the product of MgCh^{2,3}. In consequence, we and other laboratories observed the seedling-lethal phenotype of *gun4* knock-out mutant under photoperiodic growth conditions^{1,4}. And we observed that *gun4-1* (a *gun4* knock-down mutant) showed a much stronger pale-green leaf phenotype than the *bcm1* mutants do (Fig. 3a), implying that GUN4 acts decisively on the Mg chelation step, which is different from the role of BCM1.

It is nonetheless pity that authors do not provide a control assay containing recombinant BCM1 but lacking GUN4 as they do for in vitro assays containing BCM1-enriched membranes. Such experiment would show more

convincingly than current Fig. 4c and Suppl Fig. 6a whether the stimulation of Mg chelatase in vitro is provided exclusively by an interplay between BCM1 and GUN4.

Response: Since the addition of β -DM has a strong negative effect on MgCh activity in vitro (Supplementary Fig. 6b), the production of MgP was not detectable by HPLC in the assay supplemented with β -DM, but lacking GUN4. At the moment, it is difficult to analyze the effect on recombinant BCM1 on MgCh activity in the assay without GUN4. In the future, we may try to do the amphipol or nanodisc reconstitution of recombinant BCM1 and explore in more details, how BCM1 supports the organization of functional MgCh complex and controls MgCh activity at the membrane. As we suggested in the Discussion section, we are working on different aspects to further unravel the molecular mechanisms by which BCM1 antagonistically control both chlorophyll biosynthesis and breakdown.

Reviewer #2 (Remarks to the Author):

I would like to thank the authors for taking into account most of our comments. I believe this work is a significant advance in our understanding of the chl metabolism.

Response: We sincerely appreciate the time and effort in providing constructive reviewing.

However, I would still like to point out that the proposed model is somehow counter-intuitive (point 3 been made previously). I still believe that based on the data presented, BCM2 function remains somehow unclear. This is fine, given the novelty of the concepts (and the large amount of data) provided in the paper, but I would have like to have a little more in the Discussion explaining better the rationale for BCM2 function in chl degradation. There is no obvious need for a down-regulation of the one-way chl degradation pathway during leaf senescence.

Response: To understand the function of BCM2 in more perspective, more detailed time-resolved analysis of the BCMs' function is required during plant development. We suggest that BCM2 acts as a negative regulator of Chl breakdown during the transition from leaf maturation to early senescence. Both BCMs cooperatively restrict the rate of Chl degradation. In this context, it is worth to mention that deficiency of both BCMs leads to a stronger pale-green phenotype of older leaves compared to those of the *bcm1* mutant (Fig. 5a, c). Moreover, destabilization of BCM2 is suggested to enhance Chl degradation, when plants reach senescence. We complete our explanations in the Discussion part.

Reviewer #3 (Remarks to the Author):

In the revised manuscript, most of the questions had been answered. However, there are still some minor issues that need to be addressed further.

1. The quality of Fig. 6a needs to be improved. The resolution of this image is not high enough.

Response: Since the Fig. 6a is a graph with many data, we incorporated the compressed figure into the Word file of the manuscript. We have submitted the production-quality version of Fig. 6 and expect to publish this figure with high resolution.

2. I suggest the authors to change the Arabic numerals in these two paragraphs (lines 336-339; lines 398-403) to Roman numerals to make the paper more readable.

Response: We have revised these points in the manuscript.

3. *Transgenic plants should be shown in italics.*

Response: We have revised this point in the manuscript.

References

- 1 Larkin, R. M., Alonso, J. M., Ecker, J. R. & Chory, J. GUN4, a regulator of chlorophyll synthesis and intracellular signaling. *Science* **299**, 902-906, doi:10.1126/science.1079978 (2003).
- 2 Axelsson, E. *et al.* Recessiveness and dominance in barley mutants deficient in Mg-chelatase subunit D, an AAA protein involved in chlorophyll biosynthesis. *The Plant cell* **18**, 3606-3616, doi:10.1105/tpc.106.042374 (2006).
- 3 Masuda, T. Recent overview of the Mg branch of the tetrapyrrole biosynthesis leading to chlorophylls. *Photosynth Res* **96**, 121-143, doi:10.1007/s11120-008-9291-4 (2008).
- 4 Richter, A. S. *et al.* Phosphorylation of GENOMES UNCOUPLED 4 Alters Stimulation of Mg Chelatase Activity in Angiosperms. *Plant physiology* **172**, 1578-1595, doi:10.1104/pp.16.01036 (2016).